# Cytoneme-mediated cell-cell contacts for Hedgehog reception

**Laura González-Méndez, Irene Seijo-Barandiarán, Isabel Guerrero\***

Centro de Biología Molecular 'Severo Ochoa', Universidad Autónoma de Madrid, Madrid, Spain

**Abstract** Morphogens regulate tissue patterning through their distribution in concentration gradients. Emerging research establishes a role for specialized signalling filopodia, or cytonemes, in morphogen dispersion and signalling. Previously we demonstrated that Hedgehog (Hh) morphogen is transported via vesicles along cytonemes emanating from signal-producing cells to form a gradient in *Drosophila* epithelia. However, the mechanisms for signal reception and transfer are still undefined. Here, we demonstrate that cytonemes protruding from Hh-receiving cells contribute to Hh gradient formation. The canonical Hh receptor Patched is localized in these cellular protrusions and Hh reception takes place in membrane contact sites between Hh-sending and Hh-receiving cytonemes. These two sets of cytonemes have similar dynamics and both fall in two different dynamic behaviours. Furthermore, both the Hh co-receptor Interference hedgehog (Ihog) and the glypicans are critical for this cell-cell cytoneme mediated interaction. These findings suggest that the described contact sites might facilitate morphogen presentation and reception.

## Introduction

Long-distance cell-cell communication is essential for development and function of multicellular systems. During embryogenesis morphogens are produced at a localized source and act at a distance, controlling the differential activation of target genes in a concentration-dependent manner (*Crick, 1970*; *Morgan, 1901*; *Rogers and Schier, 2011*; *Stumpf, 1966*; *Turing, 1952*; *Wolpert, 1969*). Thus, the graded distribution of morphogens, together with the ability of the receptor cells to respond specifically to different ligand concentrations, need to be tightly regulated. Increasing evidence supports active signal transport along specialized filopodia (also called cytonemes) (*Ramírez-Weber and Kornberg, 1999*; *Gradilla and Guerrero, 2013a*, *2013b*; *Kornberg, 2014*), challenging models based on the free diffusion of morphogens. Here we keep investigating the mechanisms of gradient formation by active transport of morphogens across the epithelial surfaces.

The Hedgehog (Hh) morphogen has a central role in many metazoan developmental processes. Hh is implicated in stem cell maintenance, axon guidance, cell migration and oncogenesis in a wide range of organisms (*Briscoe and Thérond, 2013*). Hh production, transport, release and reception must be kept under strict spatial and temporal control to fulfil its signalling function. Hh is also post-translationally modified by the addition of cholesterol (*Porter et al., 1996*) and palmitic acid (*Pepinsky et al., 1998*), which promote its association with cell membranes making Hh transport potentially inconsistent with free morphogen diffusion. Hh gradient establishment has been extensively studied in the *Drosophila* wing imaginal disc epithelium, which is formed by anterior (A) and posterior (P) cell populations with different adhesion affinities. The P compartment cells produce Hh, which moves across the A/P compartment border to reach the Hh-responding cells in the A compartment. As Hh spreads away from the border, its concentration decreases, providing a graded signal that activates the different target genes that regulate imaginal disc development (reviewed in *Briscoe and Thérond, 2013*).

**\*For correspondence:** iguerrero@cbm.csic.es

**Competing interests:** The authors declare that no competing interests exist.

**eLife digest** When an embryo develops, it is critical that tissues and organs form properly and at the right time. For this, cells need to be able to communicate over long distances by using signalling molecules called morphogens. Morphogens disperse via extensions that protrude from the surface of a 'source' cell. Previous research has shown that these extensions called cytonemes can transport the morphogens to 'receiver' cells, and depending on the distance from the source, build a concentration gradient that will either be higher or lower. These gradients then help unspecialized cells to develop into different specialized ones.

One of the key morphogens during the development is the Hedgehog protein. Researchers have previously shown that vesicles along cytonemes of cells that produce Hedgehog transport the morphogen to the receiver cells. However, until now it was unclear how the Hedgehog signals are transferred and received.

Here, González-Méndez et al. – including researchers involved in the previous studies – investigated the cytonemes located on Hedgehog-receiving cells in the fruit fly. The results showed that these cytonemes are oriented towards the Hedgehog-producing cells and help to create a concentration gradient by varying their length. Moreover, the cytonemes from signal-producing and signal-receiving cells connect at specific sites that are distributed along their lengths. This suggests that the contact sites might help to transfer and receive the morphogens.

Thus, the way cells communicate in other tissues of the body could be similar to how nerve cells communicate with each other in the brain. Our next challenges will be to fully understand how cytonemes transfer the Hedgehog signal. This could shed more light on how Hedgehog signaling can be controlled and modulated.

In both wing disc and abdominal histoblasts, cytonemes from Hh-producing cells extend across its morphogenetic gradient (*Bischoff et al., 2013*). Critically, there is a strong correlation between the extent of cytonemes from the P compartment and the graded response to Hh signalling in the A compartment. In vivo imaging of abdominal histoblasts showed that cytonemes extend and retract dynamically, and that Hh gradient establishment correlates with cytoneme formation in both space and time. These data support a model for Hh transport in which cytonemes act as conduits for morphogen movement mainly at the basal plane of the epithelium. Furthermore, we have shown that Hh is associated with vesicles transported along cytonemes (*Gradilla et al., 2014*). The mechanisms for Hh signal transfer and reception, however, remain open questions.

Here we show that cytonemes emanating from the Hh-receiving cells in the A compartment contribute to Hh reception and gradient formation. These cytonemes have similar dynamics than those emanating from the Hh-producing cells, falling between two different dynamic behaviours. We show that reception Hh signalling components localize to the signal-receiving cytonemes, including the glypicans Division abnormally delayed (Dally) and Dally-like (Dlp), the adhesion molecule Interference hedgehog (Ihog) and the canonical Hh receptor Patched (Ptc). Significantly, the spreading capacity of cytonemes is dependent on the glypicans present in the membranes of neighbouring cells. Thus, cytonemes cannot properly extend across Dally or Dlp mutant cells. In addition, cytonemes can cross *smo* (*smoothened*) and *ptc* mutant clones, which cannot internalize Hh, providing a bridging mechanism and allowing Hh delivery to adjacent wild type cells. Finally, we describe discrete cell-cell contact structures between Hh-sending and Hh-receiving cytonemes, where the morphogen may be transferred from one cytoneme to the other for its reception.

## Results

### Hh-responding cells extend dynamic cytonemes to receive Hh

Hh-producing cells in the P compartment of the wing imaginal disc extend cytonemes that transport Hh to the A compartment cells and that are essential for the restricted distribution of Hh during *Drosophila* epithelial development (*Callejo et al., 2011*; *Bilioni et al., 2013*; *Bischoff et al., 2013*). In addition, the Hh-receiving cells of the anterior compartment also extend cytonemes towards the Hh-

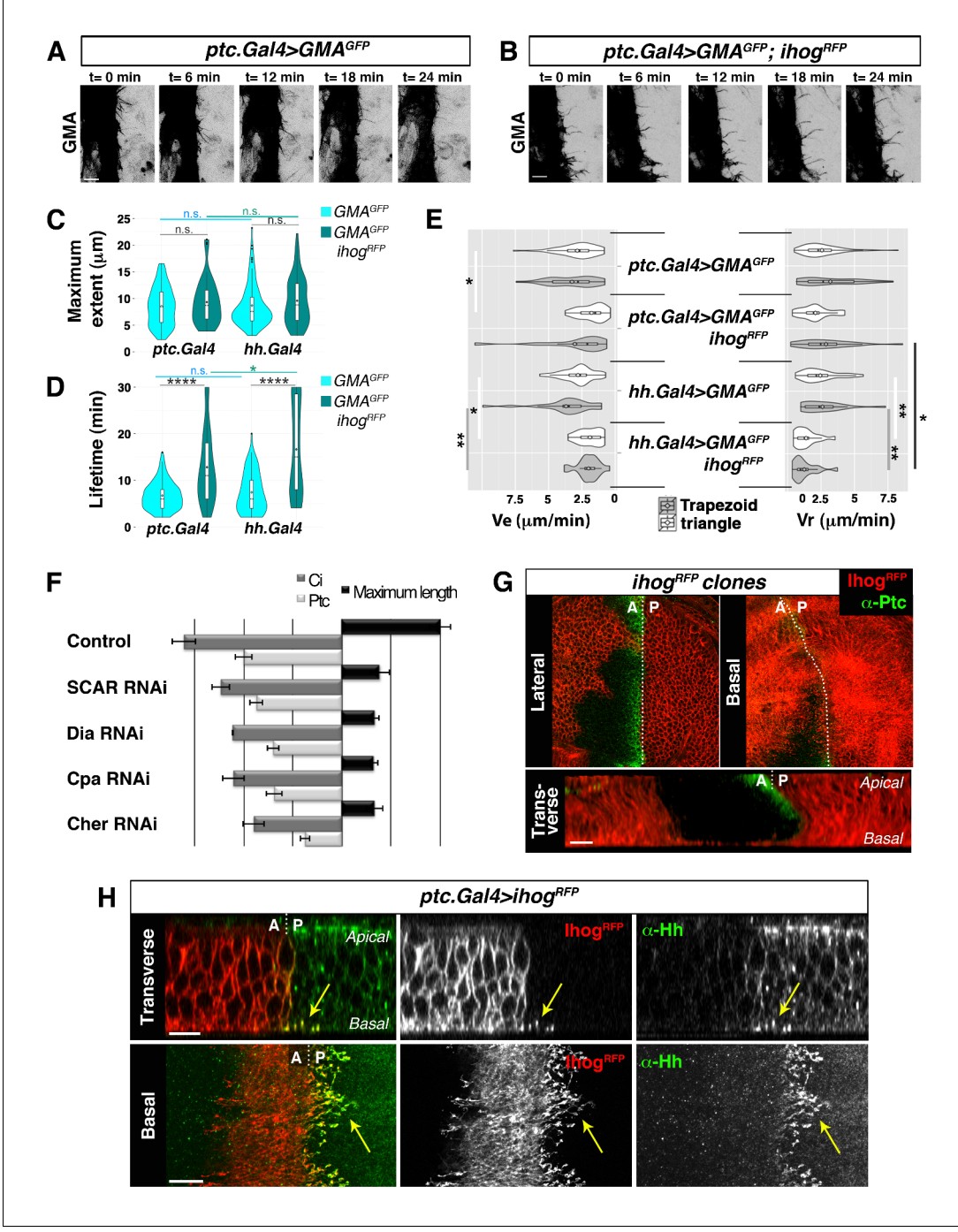

**Figure 1.** Cytonemes emanating from the A compartment cells. (**A, B**) Dynamic behaviour of cytonemes emanating from the A compartment abdominal histoblasts, monitored by the actin cytoskeleton marker GMA-GFP alone (the actin-binding domain of moesin fused to GFP) (**A**), or co-expressed with Ihog-RFP (**B**), after 24 hr of expression using the *ptc.Gal4* driver. Hh-receiving cells produce highly dynamic filopodia visualized by expressing GMA-GFP (**A**), which are more stabilized when co-expressing Ihog-RFP (**B**). (**C–E**) Violin plots represent filopodia maximum extent (**C**), lifetime (**D**), and elongation velocity (Ve) and retraction velocity (Vr) measurements (**E**). Statistical analysis was done to compare the expression, for 24 hr, of GMA-GFP alone or co-expressed with Ihog-RFP in Hh-receiving cells (*ptc.Gal4*) or Hh-producing cells (*hh.Gal4*). Coloured diamonds indicate the mean of the data and black lines the median. *p<0.05, **p<0.01, ***p<$10^{-3}$, ****p<$10^{-4}$. Scale bars represent 10 μm. (**F**) Interfering with A cytonemes extension (by transient co-expression of *UAS.ihog-RFP* together with *UAS.scar-RNAi*, *UAS.dia-RNAi*, *UAS.cpa RNAi* or *UAS.cher-RNAi* for 30 hr using the *ptc.Gal4* driver) affects the spreading of the Hh

*Figure 1 continued on next page*

*Figure 1 continued*

gradient in the wing disc, monitored by Ptc and Ci expressions (by transiently expressing the *UAS-RNAi* lines and not *UAS.ihog-RFP* for 30 hr). Graph representing the average of 5 discs in 3 independent experiments (error bars represent SDs). (**G**) Two Ihog-RFP expressing clones induced in the wing pouch, one in the A compartment and another in the P compartment, in lateral, basal and transverse (Z-stack) sections. Note that very distant cells contact at the basal region through cytonemes oriented towards the A/P compartment border (arrow). (**H**) A *ptc.Gal4, tub.Gal80<sup>ts</sup>>UAS.ihog-RFP* wing disc after 24 hr at the restrictive temperature and stained with α-Hh antibody. Note that A compartment cytonemes expressing Ihog-RFP capture Hh produced by the P compartment cells (arrows). The data shown were consistent in at least three independent experiments with an average of 5–10 discs in each experiment. Bars, 10 μm.

The following figure supplements are available for figure 1:

**Figure supplement 1.** Phase models based on filopodia dynamics.

**Figure supplement 2.** Hh-receiving cells filopodia have similar dynamics to Hh-producing cells.

**Figure supplement 3.** Ihog overexpression in Hh-receiving cells leads to an extended Hh gradient.

secreting cells of the P compartment. Here we have characterized the cytonemes from the signal-receiving cells and investigated their role in Hh morphogen reception.

In previous studies on Hh signalling filopodia in the abdominal histoblasts we showed that the P compartment generated highly dynamic protrusions that reached anteriorly the Hh-receiving cells (*Bischoff et al., 2013*). The Hh-receiving cells also produce highly dynamic protrusions oriented towards the Hh-producing cells, easily visualized when expressing the actin-binding domain of moesin (GMA) fused to GFP (*Figure 1A*, *Video 1A*). These GMA-labelled filopodia are less dynamic when they co-express Ihog (*Figure 1B*, *Video 1B*), as was previously described for the Hh-producing histoblasts (*Bischoff et al., 2013*). Here we show that both Hh-presenting and Hh-receiving histoblast cells emit protrusions with similar dynamics (*Video 1* and *Video 2*). In a more detailed analysis of filopodia dynamics, we have been able to distinguish two different dynamic behaviours: one of filopodia that elongate and immediately retract, which we have classified as 'triangle dynamics' and another one with a 'stationary' interphase between the elongation and retraction phases, which we have classified as 'trapezoid dynamics' (*Figure 1—figure supplement 1*; see Materials and methods). Both Hh-producing and Hh-receiving cell filopodia have similar values of average maximum extent, lifetime, elongation (Ve) and retraction (Vr) velocities (*Figure 1C–E*, *Figure 1—figure supplement 2*).

The maximum extent of filopodia is similar with or without overexpression of Ihog (*Figure 1C*, *Figure 1—figure supplement 2*). This overexpression, however, changes cytoneme dynamics. Filopodia lifetime values are higher for filopodia with high levels of Ihog than those with low levels (*Figure 1D*, *Figure 1—figure*

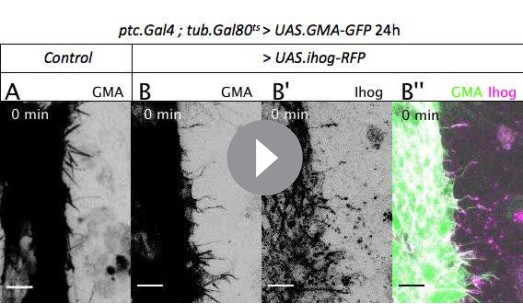

*ptc.Gal4 ; tub.Gal80<sup>ts</sup> > UAS.GMA-GFP 24h*

**Video 1.** Dynamics of filopodia of A compartment abdominal histoblast cells expressing only GMA and co-expressing Ihog. (A-B) Abdominal histoblasts of pupae with the genotype *w; ptc.Gal4 / tubGal80<sup>ts</sup>; UAS.GMA-GFP /+* (A) and *w; ptc.Gal4 / tubGal80<sup>ts</sup>; UAS.GMA-GFP / UAS.ihog-RFP* (B). The actin-binding domain of moesin fused to GFP (GMA-GFP) was expressed during 24 hr in Hh-receiver cells to visualize actin-based filopodia dynamics, easily detected using the inverted grey-scale lookup table tool of Fiji (A). Notice the highly dynamic filopodia emerging from A cells towards P cells. GMA-GFP (B) was co-expressed with Ihog-RFP (B') in Hh-receiver cells. Observe in the merge panel (B'') that Ihog-containing filopodia are stabilized while few filopodia with low or no Ihog levels detected are more dynamic. Histoblasts move up (A) and down (B) towards the dorsal midline, not shown because of the high magnification. Anterior is on the left. Pupae were around 30 hr APF (after puparium formation). Movies of 30 min imaging with time intervals between frames of 2 min. Scale bars represent 10 μm.

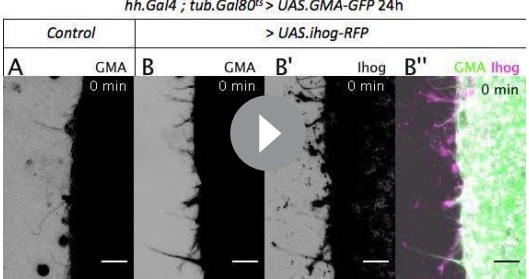

hh.Gal4 ; tub.Gal80^{ts} > UAS.GMA-GFP 24h

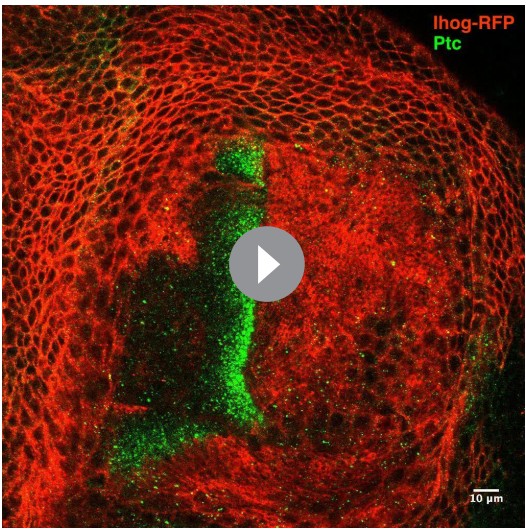

**Video 2.** Dynamics of filopodia from P compartment abdominal histoblast cells expressing only GMA and co-expressing Ihog. (A-B) Abdominal histoblasts of pupae with genotype *w; tubGal80^{ts} /+; hh.Gal4 / UAS. GMA-GFP* (A) and *w; tubGal80^{ts} / UAS.ihog-RFP; hh. Gal4 / UAS.GMA-GFP* (B). GMA-GFP was expressed during 24 hr in Hh-producing cells to visualize actin-based filopodia dynamics, easily detected using the inverted grey-scale lookup table tool of Fiji (A). Notice the highly dynamic filopodia emerging from P cells towards A cells. GMA-GFP (B) was co-expressed with Ihog-RFP (B') in Hh-producing cells. Observe in the merge panel (B'') that Ihog-containing filopodia are stabilized while few filopodia with low or no Ihog levels detected are more dynamic. Histoblasts move up (A) and down (B) towards the dorsal midline, and this is not shown because of the high magnification. Anterior is to the left. Pupae were around 30 hr APF (after puparium formation). Movies of 30 min imaging with time intervals between frames of 2 min. Scale bars represent 10 μm.

**Video 3.** Membrane contacts between A and P cell cytonemes. Z-stack, from apical to basal, of a wing disc showing the GRASP signal pattern between Hh-producing and Hh-receiving cells (*ptc.Gal4, tub. Gal80^{ts}>UAS.CD4-GFP^{1-10}/ hh.LexA>LexAop.CD4-GFP^{11}* wing disc after 24 hr at the restrictive temperature). Note the different GFP pattern in apical and basal sections.

*supplement 2*). Hh-producing and Hh-receiving cell filopodia elongate and retract at similar rates (*Figure 1E*, *Figure 1—figure supplement 2A*). In summary, Ihog overexpression leads to more stable cytonemes also in the A compartment cells.

To further analyse the role of A compartment cytonemes, we manipulated actin cytoskeletal components implicated in filopodia formation within the wing imaginal disc, monitoring the expression of Ptc and Cubitus interruptus (Ci) as a read-out of Hh signalling. We knocked down SCAR, Diaphanous (Dia), Filamin (Cheerio, Cher) or actin Capping protein alpha (Cpa) function in the A compartment using specific *RNAi* lines. In addition, we expressed these RNAis together with *ihog-RFP* to visualize cytonemes. The aberrant extension of cytonemes observed by knocking down these proteins in the Hh-receiving cells resulted in reduced Ptc and Ci expression domains (*Figure 1F*), revealing a role of the A compartment cytonemes in regulating the Hh gradient. Thus, cytonemes control both Hh export from P compartment and Hh distribution and reception in the A compartment.

Cytoneme-mediated long distant communication between Hh-producing and Hh-responding cells is visualized when Ihog-RFP marked cytonemes from the A and P compartments meet at the A/P boundary in the basal part of the epithelium (*Figure 1G*, arrow; *Video 3*). Anterior compartment cells overexpressing Ihog consistently extend basal cytonemes, which invade the P compartment and heavily accumulate Hh (*Figure 1H*,

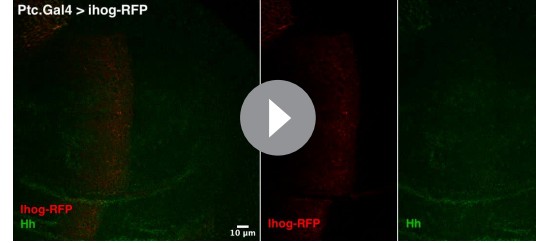

**Video 4.** Ihog overexpressing cytonemes emerging from the A compartment cells accumulate endogenous Hh produced by the P compartment cells. Z-stack, from apical to basal, of a *ptc.Gal4, tub.Gal80^{ts}>UAS.ihogRFP* wing disc after 24 hr at restrictive temperature, stained with α-Hh antibody. Note the increase of Hh levels at the basal A compartment cytonemes.

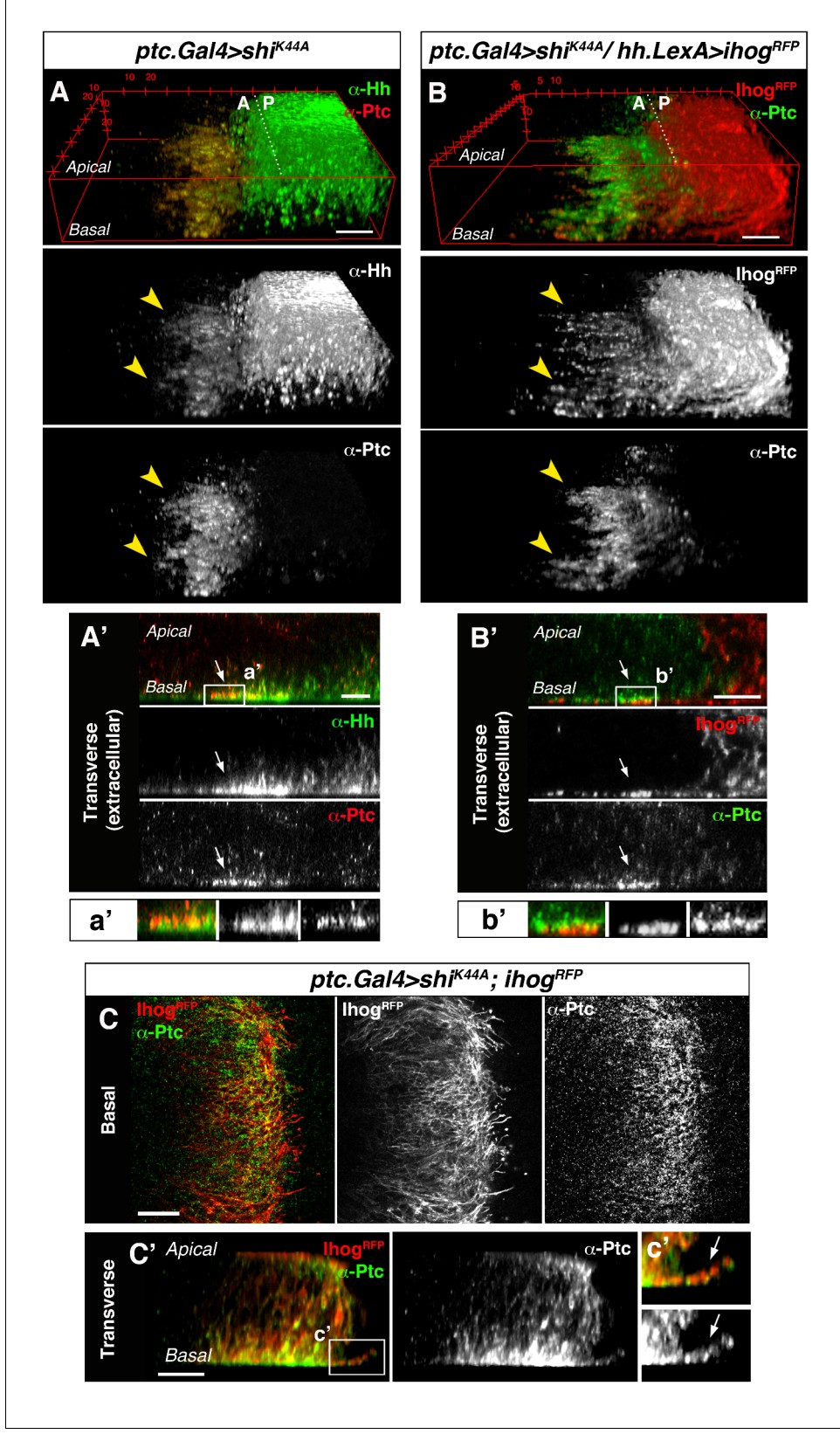

**Figure 2.** Cytonemes from A and P compartment cells interact during Hh reception. (**A**) 3D view of a *ptc.Gal4, tub. Gal80ts>UAS.shiK44A* wing disc after 12 hr at the restrictive temperature and double stained for Hh and Ptc. Note that Ptc colocalizes with Hh at the basal membranes of the Hh-receiving cells when endocytosis is frozen using a

*Figure 2 continued on next page*

*Figure 2 continued*

dominant negative form of Dynamin (arrowheads). (**A'**) Transverse section of an extracellular staining for both Hh and Ptc in a similar wing disc. Note the colocalization of Hh and Ptc in the most basal part of the disc (arrows). (**a'**) Magnification of A'. (**B**) 3D view of a *ptc.Gal4, tub.Gal80^{ts}>UAS.shi^{K44A} / hh.LexA>LexAop-ihog-RFP* wing disc after 12 hr at the restrictive temperature and labelled with α-Ptc antibody. Note Ihog-RFP localization in the P compartment cytonemes and Ptc in the A compartment cytonemes. (**B'**) Transverse section of a similar wing disc stained for extracellular Ptc. Note the colocalization of Ihog and Ptc in the most basal part of the epithelium (arrows). (**b'**) Magnification of B'. (**C, C'**) Basal (C) and transverse (C') sections of a *ptc.Gal4, tub.Gal80^{ts}>UAS.ihog-RFP>UAS.shi^{K44A}* wing disc after 12 hr at the restrictive temperature and stained with α-Ptc antibody. (**c'**) Magnification of C'. Note that when endocytosis is frozen, Ptc is located in A compartment cytonemes labelled with Ihog-RFP (arrow).

The following figure supplement is available for figure 2:

**Figure supplement 1.** Cytonemes from A and P compartment cells interact during Hh reception.

---

arrow; *Video 4*), leadingto an extended Hh gradient (*Figure 1—figure supplement 3*). Taken together, these results indicate that Hh-receiving cells generate dynamic cytonemes directed towards Hh-producing cells, and that these A cytonemes can influence the graded distribution of Hh.

## Cytonemes from producing and receiving cells interact for Hh reception

To further study the involvement of cytonemes in Hh reception we blocked endocytosis specifically in Hh-receiving cells by expressing *shi^{K44A}*, which encodes a dominant negative mutant of Dynamin (aka Shibire, Shi). Expression of this mutant protein blocks the rapid internalization of the Hh-Ptc complex in signal-receiving cells. This approach enables the visualization of ligand and receptor at the site of signal reception (*Callejo et al., 2011*; *Torroja et al., 2004*). In this mutant background, Hh co-localizes with Ptc at the basal membrane of Hh-receiving cells, revealing the presence of filopodia-like structures oriented in the A-P axis (*Figure 2A*, arrowheads). Extracellular staining of Ptc and Hh confirms this basal localization of Hh-Ptc complex for signal reception (*Figure 2A'*, arrow). Furthermore, cytonemes from Hh-producing cells (labelled with Ihog-RFP) invade the A compartment to contact the basal membrane of the Hh reception region (labelled with Ptc) in A cells (*Figure 2B* arrowheads and 2B' arrow). We can also detect the receptor Ptc localized in A cytonemes when its internalization is impeded in the Hh-receiving cells (*Figure 2C and C'*, arrow) or co-localizing with P compartment Ihog-RFP labelled cytonemes (*Figure 2—figure supplement 1*). These results indicate that A compartment cells receive the Hh signal through their cytonemes, which also contain Ptc.

We then expressed Hh-CD2 protein, which cannot be released from the plasma membrane and is therefore unable to form a concentration gradient (*Strigini and Cohen, 1997*). As expected Hh-CD2 localizes to basal cytonemes (*Figure 3—figure supplement 1A*), since this form of Hh is permanently anchored to the plasma membranes. Therefore, the Hh-CD2 protein expression allowed us to visualize interactions between cytonemes emanating from the A compartment (expressing Ihog-RFP) and cytonemes from the P compartment (expressing Hh-CD2) along their lengths (*Figure 3A*, arrowheads; *Figure 3—figure supplement 1B*). Hh-receiving cytonemes accumulate the co-receptor Ihog at contact sites with Hh-CD2-presenting cytonemes (*Figure 3—figure supplement 1C and C'*, compare C with D). Noticeably, the Hh receptor Ptc is also localized in these contact sites between cytonemes from A and P compartment cells (*Figure 3B*, arrows; *Figure 3—figure supplement 1C' and E*). These results suggest that the Hh receptor complex binds Hh-CD2 protein at the basal membrane of Hh signal-receiving cytonemes, while blockage in Hh release and internalization results in Ptc also being retained in the cytoneme membrane (*Figure 3—figure supplement 1E*, compare E with F). Taken together, these data indicate that Hh reception is mediated by contacts between Hh-sending and Hh-receiving cytonemes.

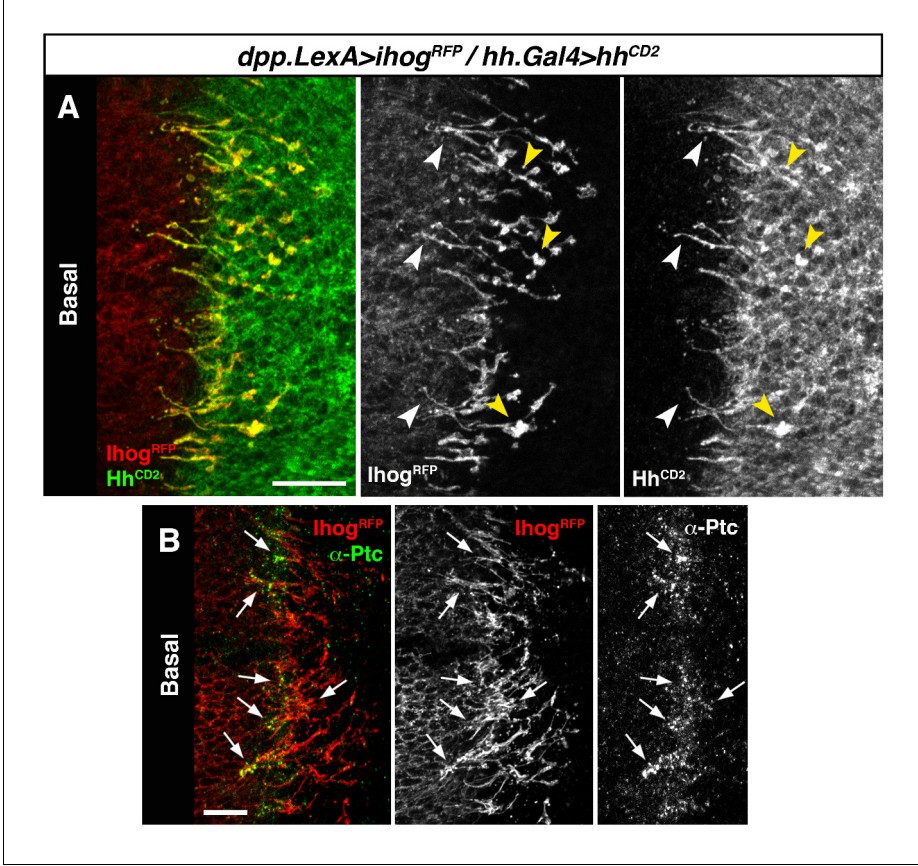

**Figure 3.** Hh reception at contacts between Hh-sending and Hh-receiving cytonemes. (**A**) A *dpp.LexA>LexAop. ihog-RFP / tub.Gal80$^{ts}$, hh.Gal4>UAS.hh-CD2* wing disc grown for 24 hr at the restrictive temperature and labelled with α-CD2 antibody. Observe that Hh-receiving and Hh-CD2-presenting cytonemes interact along their lengths. Higher levels of Ihog are in those A cytonemes that interact with P cytonemes at both sides of the A/P compartment border (white and yellow arrowheads), probably due to the higher stability of these cytonemes. (**B**) Basal section of a similar wing disc labelled with α-Ptc antibody. Observe the accumulation of Ptc and Ihog at these A cytonemes (arrows), probably because Ptc does not internalize Hh-CD2 so that the whole internalization complex is accumulated at the Hh-receiving cytonemes in the A compartment. The data shown were consistent in at least three independent experiments with an average of 5–10 discs in each experiment. Bars, 10 μm.

The following figure supplement is available for figure 3:

**Figure supplement 1.** Hh reception at contacts between Hh-sending and Hh-receiving cytonemes.

## Discrete sites for cytoneme-mediated Hh reception

To further explore the mechanism of contact between A and P cytonemes we adapted the GRASP technique (GFP Reconstitution Across Synaptic Partner), which was originally developed to image membrane contacts at neuronal synapses (*Feinberg et al., 2008*; *Gordon and Scott, 2009*). We used *UAS.CD4-GFP$^{1-10}$* with the complementary CD4-GFP$^{11}$ fragment regulated by LexAop (*LexAop.CD4-GFP$^{11}$*). Membrane contacts between Hh-sending and Hh-receiving cells were imaged by expressing the complementary GFP fragments separately in each compartment using the required LexA and Gal4 drivers. As expected, GFP fluorescence was localized to the A/P border when complementary fragments were expressed at the same time in each compartment. The pattern of GFP fluorescence varies along the apico-basal axis, as both A and P cells extend cytonemes along the basal surface of the epithelium (*Figure 4A,B,B'*; *Video 5*). Cytoneme membrane contacts occur on both sides of the A/P compartment border, although predominantly on the A side (*Figure 4C*). Furthermore, periodic annular structures are present along the length of overlapping cytoneme (arrows

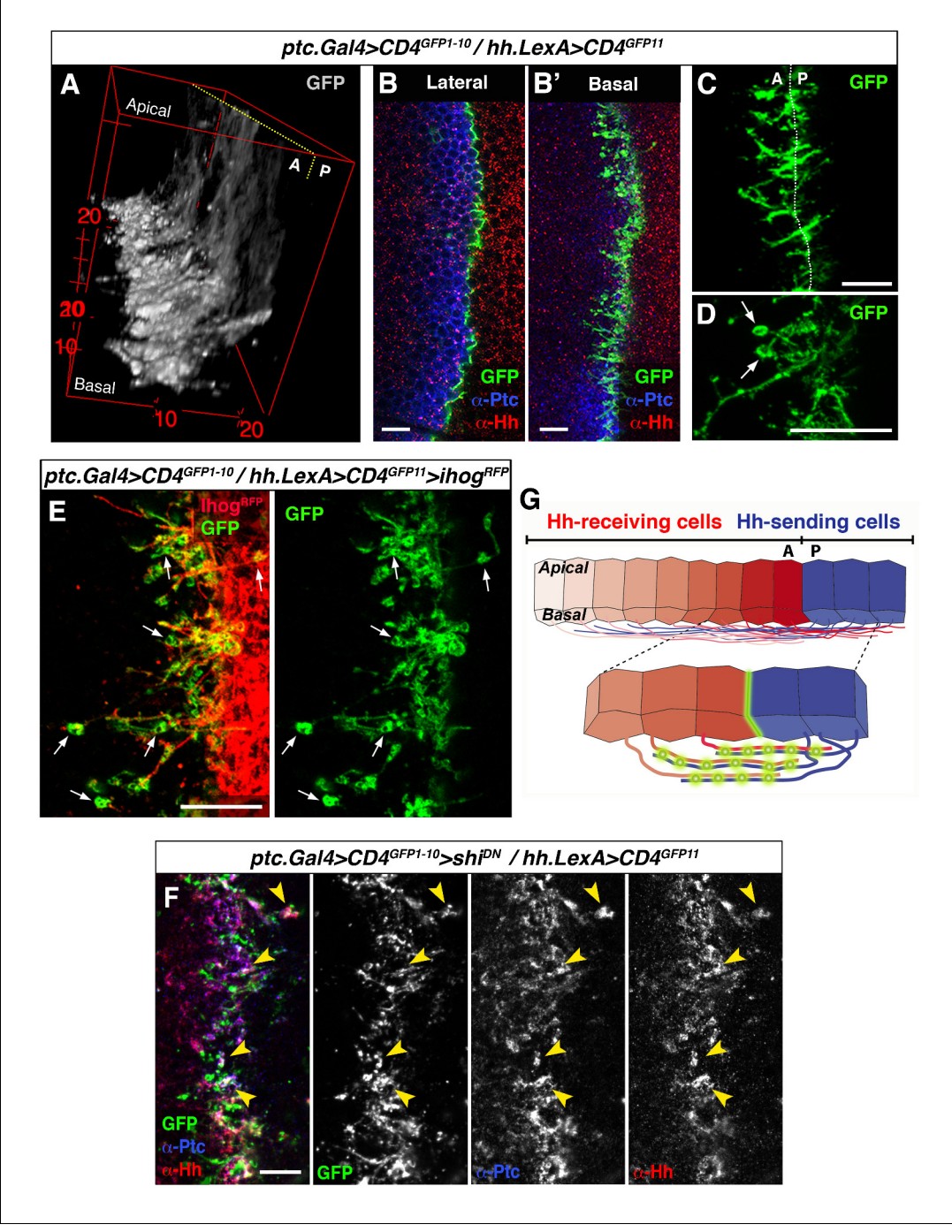

**Figure 4.** GRASP showing cytoneme-cytoneme interaction at the A/P compartment border. (**A**) 3D view of a *ptc. Gal4, tub.Gal80^{ts}>UAS.CD4-GFP^{1-10} / hh.LexA>LexAop.CD4-GFP^{11}* wing disc after 24 hr at the restrictive temperature. (**B, B'**) A wing disc of the same genotype as in A co-labelled with α-Ptc and α-Hh antibodies. Note the GFP complementation at the A/P compartment border in a lateral (**B**) and a basal section (**B'**). (**C, D**) Basal sections showing GRASP fluorescence in similar wing discs. Cytonemes cross the A/P compartment border from A to P and from P to A (**C**). Annular rings are visualized along these interacting cytonemes (**D**, arrows). (**E**) A *ptc.Gal4, tub.Gal80^{ts}>UAS.CD4-GFP^{1-10} / hh.LexA>LexAop.CD4GFP^{11}>LexAop.ihogRFP* wing disc after 24 hr at the restrictive temperature to visualize Ihog labelled cytonemes emanating from the P compartment. Note the GRASP signal in the circular structures attached to cytonemes (arrows). (**F**) A *ptc.Gal4, tub.Gal80^{ts}>UAS.CD4-GFP^{1-10}>UAS.shi^{K44A} / hh.LexA>LexAop.CD4-GFP^{11}* wing disc after 12 hr at the restrictive temperature and co-labelled with α-Hh and α-Ptc antibodies. Note the colocalization of GRASP, Ptc and Hh in the same structures at the most

*Figure 4 continued on next page*

*Figure 4 continued*

basal part of the disc (arrowheads). (**G**) Diagram depicting cytoneme interactions at the A/P compartment border. The green colour corresponds to the GRASP signal at the A/P compartment border and also at the specific sites for Hh reception along overlapping cytonemes. The data shown were consistent in at least four independent experiments with an average of 8 discs in each experiment. Bars, 10 µm.

in *Figure 4D and E*), while the GRASP signal also indicates that cytonemes overexpressing Ihog-RFP are indeed in contact with neighbouring cell membranes, presumably receiving cytonemes (*Figure 4E*, arrows). Importantly, both Hh and Ptc co-localize in these basal membrane contacts (detected by GRASP signal) upon blockage of their uptake in a $shi^{K44A}$ mutant background (*Figure 4F*, arrowheads). Therefore, we propose that these contacts are sites where Hh-Ptc interaction takes place all along the length of A and P cytonemes for signal reception (see diagram *Figure 4G*).

## Cytonemes bridge $ptc^{-/-}$ and $smo^{-/-}$ mutant clones to re-establish Hh reception in adjacent territories

It has been previously described that clones of $ptc^{-/-}$ and $smo^{-/-}$ located at the A/P compartment border are unable to internalize Hh, but do not block Hh signal transmission to more anterior wild type cells (*Chen and Struhl, 1996*). Our cytoneme transport hypothesis predicts that Hh could be transported across mutant $ptc^{-/-}$ and $smo^{-/-}$ clones in cytonemes that act as conveyer belts for long distance transport. Accordingly, we observed that Hh-producing cells still spread cytonemes when Hh reception was prevented in the A compartment by knocking down either Ptc or Smo, or both co-receptors Ihog and Brother of Ihog (Boi) at the same time (*Figure 5—figure supplement 1*). In addition, cytonemes from both compartments were still formed upon the elimination of Hh (*Figure 5—figure supplement 2A–F*) using the *hh* thermosensitive allele ($hh^{ts2}$) (*Ma et al., 1993*). No apparent changes in Hh-receiving cytonemes were also observed when Hh reception was prevented in $ptc^{-/-}$ MARCM clones expressing Ihog-YFP (*Figure 5—figure supplement 2G*). Therefore, the presence of the network of cytonemes is independent of Hh signaling.

We then asked if cytonemes could facilitate the Hh transport across a *ptc* mutant territory. Hh signalling is fully activated in $ptc^{-/-}$ mutant clones due to the lack of normal Ptc repression upon Smo (*Chen and Struhl, 1996*). We show here that Hh protein can be detected in the wild type cells anterior to the $ptc^{-/-}$ mutant clone (*Figure 5A*). Accordingly, with previous experiments (*Figure 5—figure supplement 1B*), cytonemes emanating from Hh-producing cells orient

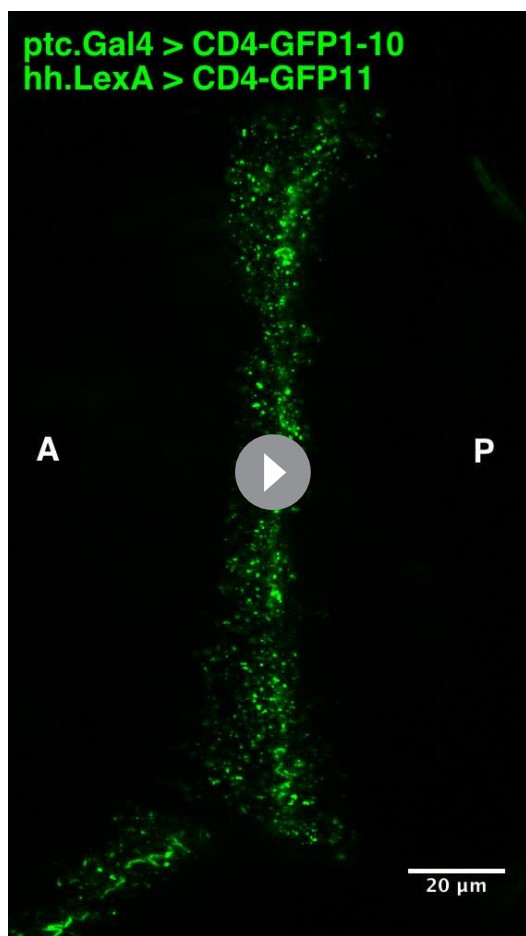

**Video 5.** Membrane contacts between A and P compartment cells. Z-stack, from apical to basal, of a wing disc showing the GRASP signal pattern between Hh-producing and Hh-receiving cells (of a *ptc.Gal4, tub.Gal80^{ts}>UAS.CD4-GFP^{1-10}/ hh.LexA>LexAop.CD4-GFP^{11}* wing disc after 24 hr at the restrictive temperature). Note the different pattern in apical and basal sections.

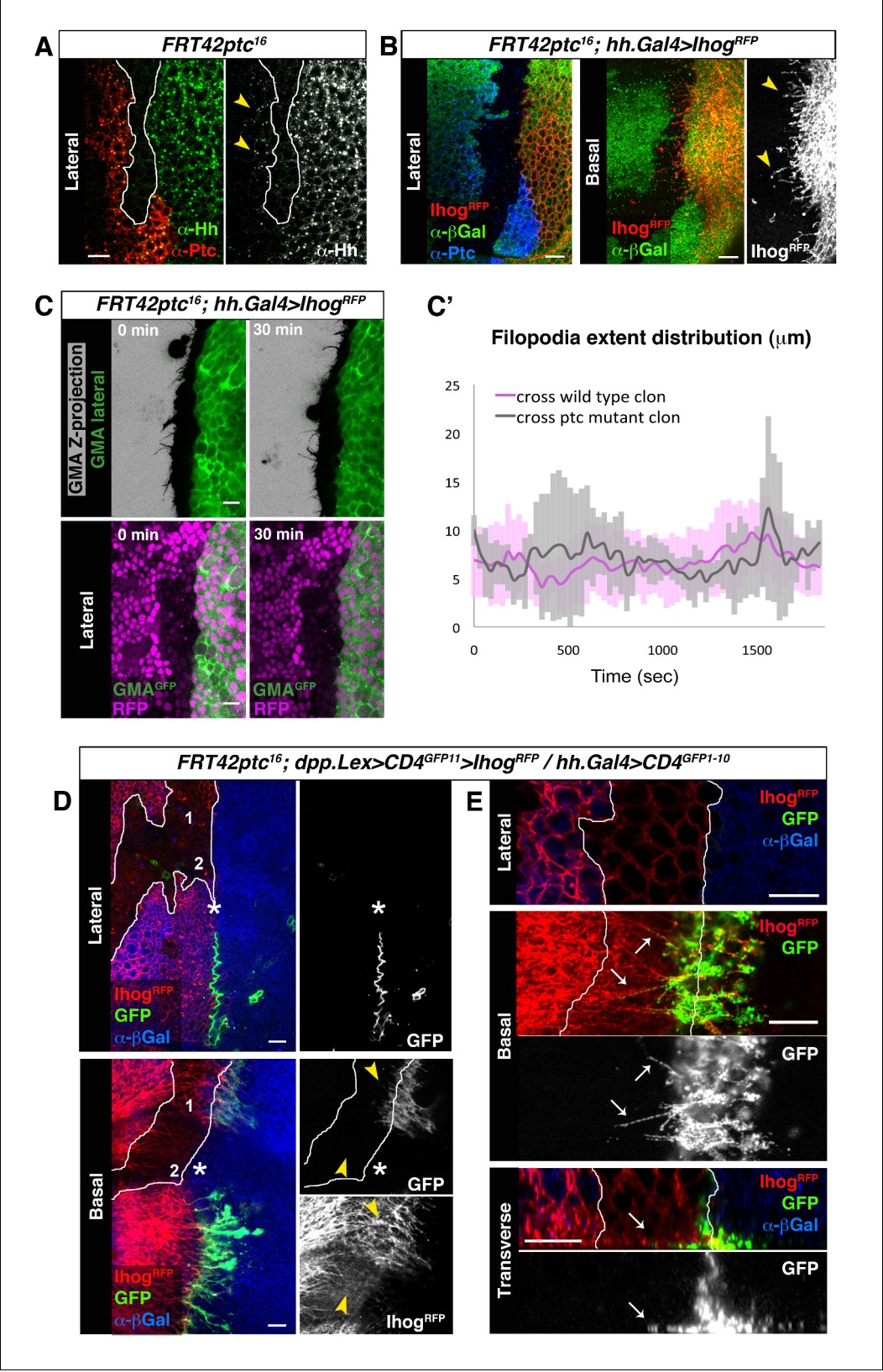

**Figure 5.** Cytonemes cross $ptc^{-/-}$ mutant clones at the A/P compartment border. (**A**) A $ptc^{16}$ null clone in the A compartment abutting the A/P compartment border of a wing disc co-labelled with α-Hh and α-Ptc antibodies. Note that both Ptc and Hh proteins are detected anterior to the clone in the A compartment (arrowheads). (**B**) P

*Figure 5 continued on next page*

*Figure 5 continued*

compartment cytonemes extend through a *ptc*$^{-/-}$ clone (arrowheads). Lateral and basal sections of a *ptc*$^{-/-}$ clone (absence of $\beta$Gal) in the A compartment with Ihog-RFP expression in the P compartment to visualize cytonemes (*FRT42D ptc*$^{16}$ / *hh.Gal4, tub.Gal80*$^{ts}$>*UAS.ihog-RFP* wing disc after 24 hr at the restrictive temperature) co-labelled with $\alpha$-$\beta$Gal and $\alpha$-Ptc antibodies. (C) First and last time frames from *Video 6* displaying a lateral view of GMA-GFP signal to easily visualize cell perimeters together with a Z-projection of GMA-GFP where filopodia are shown (top panels) or with a lateral view of nuclear RFP to distinguish between wild-type (magenta nuclei) and *ptc*$^{-/-}$ mutant (absence of magenta nuclei) territories (bottom panels). Scale bars represent 10 $\mu$m. (C') Graph showing extent distribution over time of GMA-GFP filopodia emanating from Hh-producing cells. Notice that there is no difference between filopodia crossing a wild-type (magenta) or a *ptc*$^{-/-}$ mutant (grey) clone territories. (D) A *ptc*$^{16}$ clone induced in the A compartment abutting the compartment border in a *dpp.LexA*>*LexAop.CD4-GFP*$^{11}$>*LexAop.ihog-RFP* / *tub.Gal80*$^{ts}$, *hh.Gal4*>*UAS.CD4-GFP*$^{1-10}$ wing disc after 24 hr at the restrictive temperature and labelled with $\alpha$-$\beta$Gal antibody to identify the mutant clone (absence of $\beta$Gal). Note the GRASP signal is not visualized laterally (asterisk). Note also that basal cytonemes cross the *ptc*$^{-/-}$ clone in region 1 but not in region 2, and that the GRASP signal is restricted to region 1 and absent from region 2 (arrowheads). E) Another *ptc*$^{-/-}$ clone induced in a disc with the same genotype as in D showing the interaction between cytonemes from wild type cells anterior to the clone and cytonemes from the P compartment. Note the GRASP signal along basal cytonemes from wild type A compartment cells that traverse the *ptc*$^{-/-}$ clone (arrows). The data shown were consistent in at least three independent experiments with an average of 5–10 discs in each experiment. Bars, 10 $\mu$m.

The following figure supplements are available for figure 5:

**Figure supplement 1.** Cytonemes from Hh-producing cells are still formed in mutants that modify Hh reception.

**Figure supplement 2.** Cytonemes are still present in absence of Hh ligand.

**Figure supplement 3.** Manual tracking of filopodia arising from Hh-producing abdominal histoblasts cells crossing either a *ptc* mutant or a wild-type territories.

normally towards the A compartment cells across a *ptc* mutant clone induced adjacent to the A/P border in the wing disc (*Figure 5B*). Consistently, abdominal histoblasts of the P compartment produce cytonemes that cross a *ptc*$^{-/-}$ clone in their way towards the anterior cells (*Figure 5C*, *Video 6*). These cytonemes have similar dynamics (maximum extent, lifetime and elongation and retraction velocities) when crossing wild type or *ptc*$^{-/-}$ mutant territories (*Figure 5C and C'*, *Figure 5—figure supplement 3*). Therefore, cytoneme dynamics is independent of the presence of the receptor Ptc.

In addition, GRASP signal indicates that cytonemes from the Hh-receiving cells reach the Hh-sending cytonemes close to the A/P compartment border (*Figure 5D*, region 1 arrowhead). So that, GRASP signal is detected along these A cytonemes that cross the clone and interact with the P compartment cytonemes (*Figure 5E*, arrows). However, in subapical regions of the epithelium, where cytonemes are too short to reach the P compartment, GRASP fluorescence is not detected (*Figure 5D*, asterisk). It is also known that when a *ptc*$^{-/-}$ clone is too large, wild type anterior cells are not able to respond to Hh (*Chen and Struhl, 1996*). Noticeably, in this situation we observe that cytonemes extending from the A cells do not reach the P cells and no GRASP fluorescence is detected (*Figure 5D*, region 2 arrowhead).

In *smo*$^{-/-}$ clones originating in the A compartment, and monitored by low levels of Ci expression, the A/P compartment border is re-established in the position where Hh signalling is activated in cells anterior to the clone (*Blair and Ralston, 1997*; *Rodriguez and Basler, 1997*). As a result, A compartment *smo*$^{-/-}$ cells integrate into the P compartment and maintain a straight boundary at the nearest anterior cells where Hh is then received. The Hh visualized within *smo*$^{-/-}$ clones is restricted to the most basal region of epithelial cells, demonstrating again that Hh travels mostly basally (*Figure 6A*, arrowhead) (*Bilioni et al., 2013*; *Bischoff et al., 2013*; *Callejo et al., 2011*). In addition, cytonemes emanating from cells anterior to the clone are loaded with Hh (*Figure 6B*, arrowhead), and P cytonemes cross a *smo*$^{-/-}$ clone reaching the A/P compartment border (*Figure 6C*, arrow).

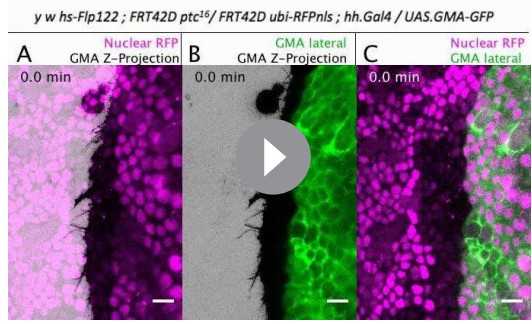

y w hs-Flp122 ; FRT42D ptc¹⁶/ FRT42D ubi-RFPnls ; hh.Gal4 / UAS.GMA-GFP

**Video 6.** Cytonemes from the P compartment abdominal histoblasts are able to cross *ptc* mutant clones and their dynamics do not differ from those crossing wild-type territories. (A-C) Abdominal histoblasts of a pupa with the genotype *y w hs-Flipase122; FRT42D ptc¹⁶ / FRT42D ubiRFPnls; hh.Gal4 / UAS.GMA-GFP*. GMA-GFP is expressed during all development at 18°C and *ptc¹⁶* clones were induced doing a heat-shock of 1 hr at 37°C two days before imaging. (A) Merge of nuclear RFP (magenta) and a Z-projection of GMA-GFP (inverted grey-scale lookup table). Hh-producing cell cytonemes labelled with the actin marker GMA (inverted grey-scale lookup table) have the same dynamics and cross normally wild-type (nuclear RFP, magenta) and *ptc¹⁶* mutant clone (absence of magenta) territories. (B) Merge of the Z-projection of GMA (inverted grey-scale lookup table) and the lateral side (green) to show the morphology of the epithelium. (C) Merge of nuclear RFP (magenta) and GMA in a lateral view (green). Here we visualize the *ptc¹⁶* mutant clone (absence of magenta) anterior to the A/P boundary. Histoblasts move down towards the dorsal midline, and this is not shown because of the high magnification. Anterior is on the left. Pupa is around 30 hr APF (after puparium formation). Movie of 30 min imaging with a time interval between frames of 30 s. Scale bars represent 10 µm.

Accordingly, GRASP fluorescence is detected where the compartment border is re-established anterior to a *smo⁻/⁻* clone (*Figure 6D*, arrow).

All together these results indicate that Hh can be transported and received by cytonemes from P and A compartment, respectively, to cross *ptc⁻/⁻* and *smo⁻/⁻* mutant territories and re-establish the morphogen gradient.

## Glypicans/Ihog trans interaction facilitate cytoneme mediated cell-cell contacts

Heparan sulphate proteoglycans (HSPGs) are key players in cell-matrix interaction, cell-cell signalling, and long-range Hh function (*Lin, 2004*; *Tabata and Takei, 2004*). We have previously demonstrated that P compartment cytonemes do not normally extend along mutant HSPG clones located at the A/P compartment border (*Bischoff et al., 2013*). Here we analyse the potential role of HSPG glypicans, in cytoneme function. The *Drosophila* glypicans Dally and Dlp are required for both Hh presentation and reception (*Ayers et al., 2010*; *Bilioni et al., 2013*; *Han et al., 2004a*; *Yan et al., 2010*). We observe that extension of Ihog-RFP expressing cytonemes is significantly reduced when glypicans levels are lowered by knocking down Dally or Dlp, in the opposite compartment (*Figure 7A–C*, arrows; *Figure 7—figure supplement 1A–D*, arrows). In agreement with this observation, cytonemes are rarely detected crossing through *dally* and *dlp* double mutant clones (*Figure 7D*, arrowhead). In addition, cytonemes emanating from the A compartment cells accumulate Dally and Dlp present in the neighbouring P compartment cells (*Figure 7E*, arrow; *Bilioni et al., 2013*). This effect is observed by expressing Ihog-RFP and eliminating Dlp in the same compartment; Dlp of one compartment accumulate in labelled cytonemes emanating from other one (*Figure 7F*, arrow). This dependence on the glypicans present in the neighbouring cells is also observed for the P compartment cytonemes (*Figure 7—figure supplement 1E and F*, arrows). In summary, all these results demonstrate the requirement of glypicans for cytoneme spreading and their specific role in the cytoneme-mediated interplay between Hh-producing and Hh-receiving cells.

We then analysed the interaction between confronting cytonemes at the A/P compartment border when Ihog is overexpressed in one compartment and either Dally or Dlp overexpressed in the other one. Remarkably, we observe an accumulation of Ihog and glypicans along the membranes of the Hh-sending and Hh-receiving interacting cytonemes (*Figure 8A–D*, arrowheads). Moreover, these contact sites present well-defined annular structures containing both Ihog and the glypicans Dally and Dlp (*Figure 8A′ and B′*, arrows) similar to the ones visualized by GRASP (*Figure 4D and E*, arrows). These results show again that Hh-sending and Hh-receiving cytonemes touch-contact along their lengths and, importantly, reveal a role for an Ihog/glypican interaction in trans during this cell-cell interface.

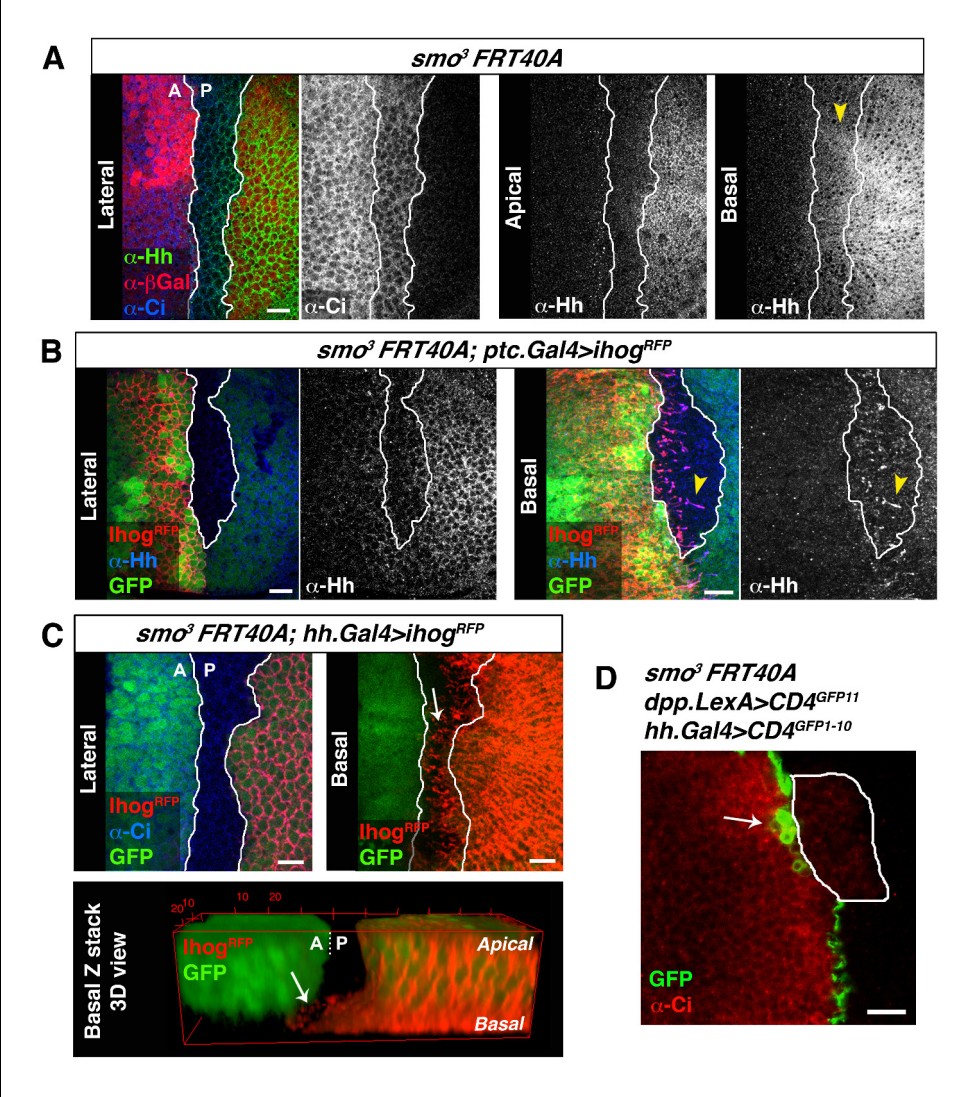

**Figure 6.** Cytonemes cross *smo*$^{-/-}$ mutant clones located at the A/P compartment border. (**A**) A *smo*$^{3}$ mutant clone in a wing disc co-labelled with α-Hh, α-Ci and α-*β*Gal to mark the clone (absence of *β*Gal). The *smo*$^{-/-}$ clone has an A compartment origin as it expresses low levels of Ci and shows no Hh localization in apical section. Interestingly, in a basal section of this disc Hh is visualized in Ci expressing cells (arrowhead), indicating that Hh moves basally and not apically through a *smo*$^{3}$ mutant clone originated in the A compartment. (**B**) A *smo*$^{3}$ mutant clone induced in a wing disc that also expresses Ihog-RFP (*smo*$^{3}$ *FRT40 / ptc.Gal4, tub.Gal80*$^{ts}$*>UAS.ihog-RFP*) after 24 hr at the restrictive temperature to visualize cytonemes emanating from cells anterior to the clone. Note that Hh is present in cytonemes from cells located anterior to the clone (arrowheads). (**C**) A *smo*$^{3}$ mutant clone (absence of GFP) induced in the A compartment abutting the A/P compartment border in a disc expressing *ihog-RFP* in the P compartment (*smo*$^{3}$ *FRT40A / hh.Gal4, tub.Gal80*$^{ts}$*>UAS.ihog-RFP* wing disc after 24 hr at restrictive temperature before dissection). Note that cytonemes emanating from P compartment cells cross along the *smo*$^{-/-}$ mutant clone (arrows). (**D**) A *smo*$^{3}$ clone of A compartment origin, identified by low levels of Ci expression, induced in a background to visualize the GRASP signal (*smo*$^{3}$ *FRT40A / ptc.Gal4, tub.Gal80*$^{t}$ *>UAS.CD4-GFP*$^{1-10}$ / *hh.LexA>LexAop.CD4-GFP*$^{11}$ wing disc after 24 hr at the restrictive temperature). Note that cells of the clone are integrated in the P compartment and the GRASP signal is located anterior to the *smo*$^{-/-}$ clone (arrow). The data shown were consistent in at least three independent experiments with an average of 5–10 discs in each experiment. Bars, 10 μm.

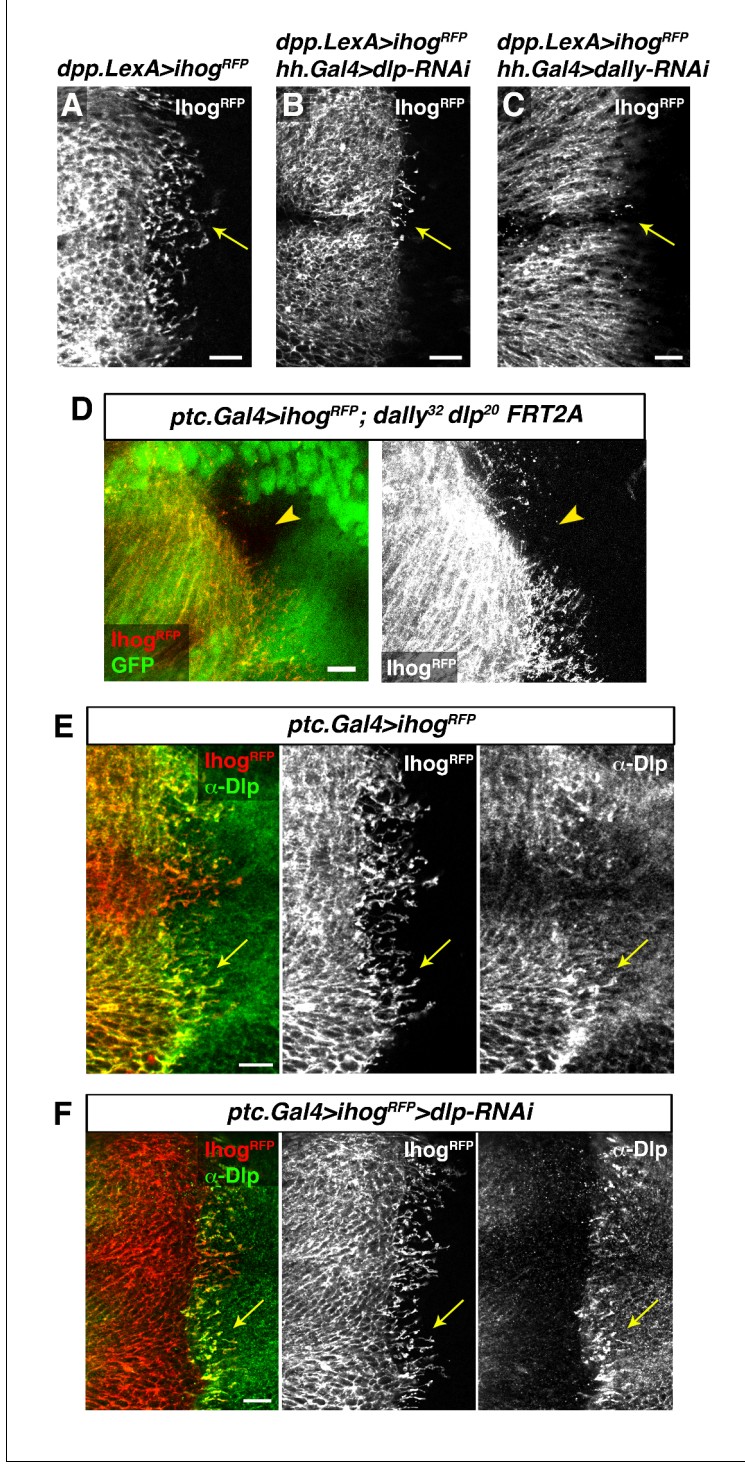

**Figure 7.** Interaction with glypicans is required for cytoneme stabilization by Ihog in Hh-receiving cells. (**A, B, C**) Ihog-RFP labelled cytonemes arising from the A compartment cells (A, *dpp.LexA>LexAop.ihog-RFP / tub.Gal80^ts*) are dependent on the glypicans Dlp (B, *dpp.LexA>LexAop.ihog-RFP / tub.Gal80^ts, hh.Gal4>UAS.dlp-RNAi*) and Dally (C, *dpp.LexA>LexAop.ihog-RFP / tub.Gal80^ts, hh.Gal4>UAS.dally-RNAi*) levels in the P compartment (arrows). All larvae were grown 30 hr at the restrictive temperature before dissection. (**D**) A *dally^32 dlp^20 FRT2A* double mutant clone (absence of GFP) induced in the P compartment and touching the A/P compartment border in a wing disc that expresses *ihog-RFP* in the receiving cells (*ptc.Gal4, tub.Gal80^ts>UAS.ihog-RFP*) to visualize cytonemes. Note the loss of cytoneme visualization crossing the clone (arrowhead). (**E**) Endogenous Dlp is accumulated in A compartment cytonemes expressing Ihog-RFP (*ptc.Gal4, tub.Gal80^ts>UAS.ihog-RFP*) after 30 hr

*Figure 7 continued on next page*

*Figure 7 continued*

at the restrictive temperature (arrows). (**F**) Wing disc showing that the endogenous Dlp accumulated in A compartment cytonemes belongs to the P compartment cells (arrows), since Dlp has been knocked down in the A compartment (*ptc.Gal4, tub.Gal80^{ts}>UAS.ihog-RFP>UAS-dlp-RNAi*) after 30 hr at the restrictive temperature before dissection. The data shown were consistent in at least three independent experiments with an average of 5–10 discs in each experiment. Bars, 10 μm.

The following figure supplement is available for figure 7:

**Figure supplement 1.** Trans interaction with glycans is required for cytoneme stabilization by Ihog in Hh-producing cells.

## Discussion

### Hh-receiving cells cytonemes contribute to Hh gradient formation

In a previous study we described that the P compartment cytonemes were critical for the Hh gradient establishment acting as conduits for Hh transport (*Bischoff et al., 2013*). Here we have extended this work and showed that the cytonemes emanating from signal-receiving cells in the A compartment also have a critical role in Hh gradient formation. Therefore, both A and P compartment cytonemes weigh in the formation of the Hh gradient. The two sets of cytonemes make contact through the basal side of the epithelium and show similar values of average maximum extent, lifetime, and elongation and retraction velocities. A and P cytonemes contact predominantly on the anterior side of the compartment boundary. Furthermore, receiving cytonemes include components of the Hh reception complex (Ptc, Ihog, Dlp and Dally), supporting their signalling role; consistently, we find that manipulating their actin dynamics modifies Hh graded signalling.

As implied in our earlier study, signal transmission might be dependent on cytoneme stability and the frequency with which A and P cytonemes make contact. More Hh would be delivered to A cells close to the A/P compartment border than to those further from the signal source. In agreement with this hypothesis, GRASP fluorescence analysis suggests that A cytonemes from cells close to the A/P boundary establish contact primarily with the cytonemes from adjacent P compartment cells, where Hh would also be delivered by further P cells cytonemes. The sum of these spatial interactions would then result in high levels of Hh signalling close to the A/P border. The longest cytonemes from cells further from the A/P compartment border would interconnect to deliver the reduced levels of Hh signalling in more distant A cells.

### Hh pathway components are crucial for A and P compartment cytoneme interaction

We analysed the behaviour of confronting A and P cytonemes with mutated or overexpressed Hh signalling components. Interestingly, the expression in the P compartment of membrane-tethered Hh (Hh-CD2), which results in a non graded signalling response (*Strigini and Cohen, 1997*), alters the receiving cytonemes cargo by hindering Ptc internalization. Despite that Hh receptor Ptc is usually rapidly internalized and not visualized at the plasma membrane, both Ptc and Ihog accumulate in A compartment cytonemes at contact sites with Hh-CD2 expressing P compartment cytonemes. Thus, this form of Hh attached to membranes plausibly affects the reception process, potentially explaining the absence of a signalling gradient and further confirming that Hh release from the presenting membranes is necessary. The mechanism by which Hh is liberated from plasma membrane anchoring lipids in *Drosophila* remains unknown. However, in strong support of a Hh shedding process, proteolytic removal of both lipidated peptides releases active Sonic hedgehog (Shh) from the surface of cultured cells (*Ohlig et al., 2011*).

On the other hand, Ihog trans interaction with glycans in cytoneme-mediated cell-cell contacts has a critical role for cytoneme dynamics. The spreading capacity of Ihog overexpressing cytonemes is dependent on the glycans presence in the membranes of neighbouring cells. Cytonemes protruding from one compartment aberrantly extend over the other compartment when is mutant for *dally* and *dlp*. This is in agreement with our previous results showing that the Tout-velu (Ttv) and Brother of Ttv (Btv) proteins (needed for the HSPGs synthesis) are required to stabilize Ihog

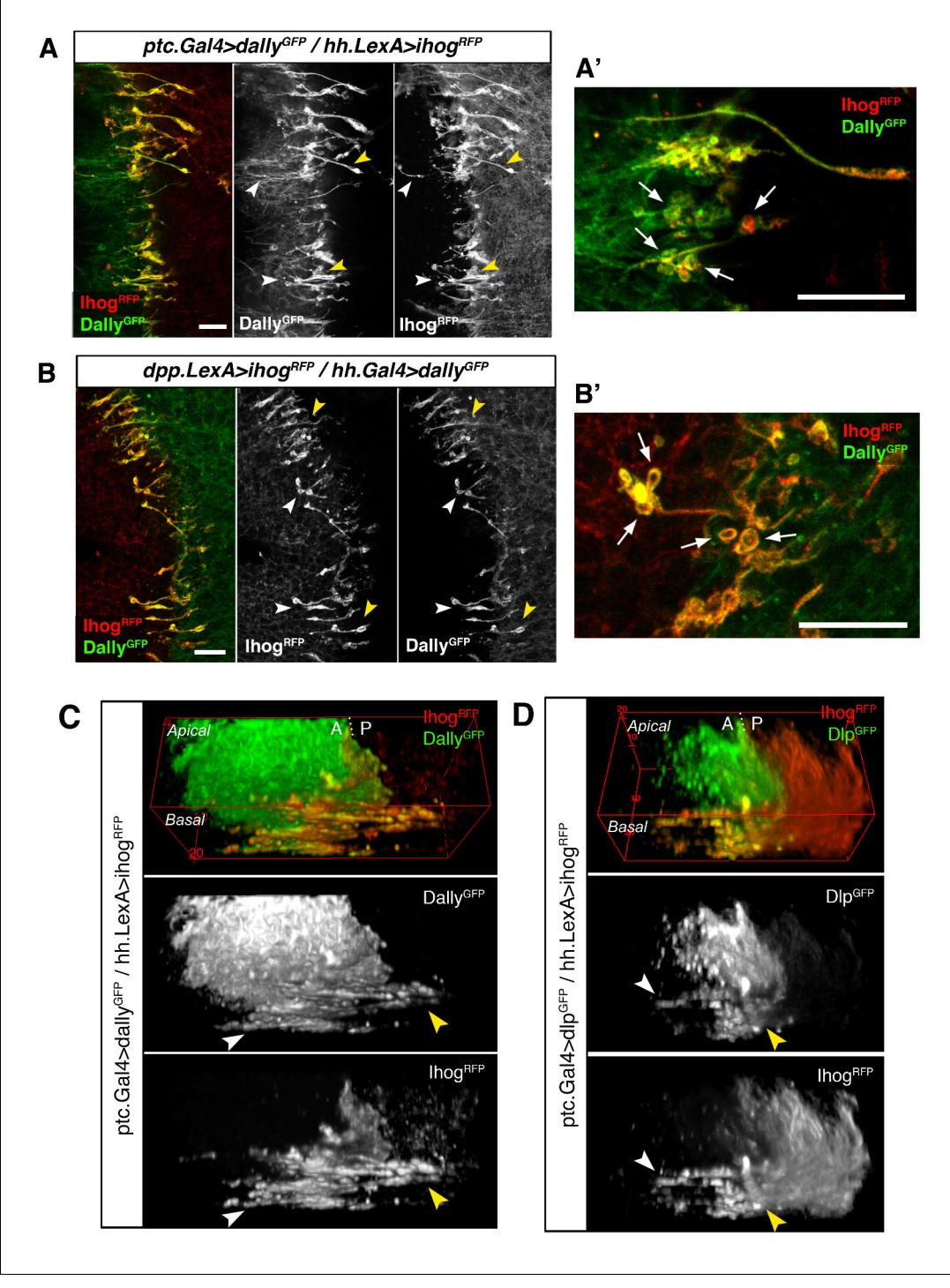

**Figure 8.** Cytonemes from A and P compartments contact by Ihog-glypicans trans interaction. (**A**) Basal P compartment cytonemes expressing Ihog-RFP interact with A compartment cytonemes expressing Dally-GFP (*ptc. Gal4, tub.Gal80$^{ts}$>UAS.dally-GFP / hh.LexA>LexAop.ihog-RFP*) at both sides of the A/P compartment border (white and yellow arrowheads). (**A'**) Enlargement of a similar disc showing annular rings, labelled by Dally-GFP and Ihog-RFP, associated with interacting cytonemes (arrows). (**B**) Basal A compartment cytonemes expressing Ihog-RFP interact with P compartment cytonemes expressing Dally-GFP (*dpp.LexA>LexAop.ihog-RFP / tub.Gal80$^{ts}$, hh. Gal4>UAS.dally-GFP*) at both sides of the A/P compartment border (white and yellow arrowheads). (**B'**) Enlargement of a similar disc showing ring structures associated with interacting cytonemes, labelled by Dally-GFP and Ihog-RFP (arrows). (**C**) 3D view of a similar wing disc showing this interaction between basal A and P
*Figure 8 continued on next page*

*Figure 8 continued*
cytonemes at both sides of the A/P compartment border (white and yellow arrowheads). (**D**) 3D view of a *ptc.Gal4,*
*tub.Gal80^ts >UAS.dlp-GFP / hh.LexA>LexAop.ihog-RFP* wing disc. P compartment cytonemes expressing Ihog-
RFP interact with A compartment cytonemes expressing Dlp-GFP at both sides of the A/P border (white and
yellow arrowheads). The larvae from A-D were grown for 24 hr at the restrictive temperature before dissection. The
data shown (**A–D**) were consistent in at least three independent experiments with an average of 5 discs in each
experiment. Bars, 10 µm.

cytonemes (*Bischoff et al., 2013*). In addition, Dally or Dlp overexpressing cytonemes from one compartment seem to be stabilized by Ihog overexpressing cytonemes from the other one, with which they associate along their length by trans interaction between Ihog and glypicans. Thus, cytonemes from one compartment can contribute to the stability of the cytonemes from the other one with which they contact.

However, to elucidate the specific mechanism of this altered cytoneme behaviour (impairment in elongation, stability or retraction processes) further in vivo analysis is required.

Taken together, these results establish that dynamic interactions between A and P compartment cytonemes play an important role in the formation of the Hh gradient. In conjunction components for both reception and presentation of Hh critically influence the performance of this cytoneme interaction.

## Hh-reception takes place at sites of direct contact between opposing A and P cytonemes

The membrane-bound GRASP fluorescence shows that A and P compartment cytonemes overlap in a region spanning the A/P compartment boundary. Within this region, direct 'kissing' contacts appear along the membranes of interacting A and P cytonemes to connect the two sets of cytonemes. These contact points are associated with annular structures, which may represent more stable links and are reminiscent of synaptic buttons. Similar 'synaptic contacts' have been previously described between air sac primordium (ASP) and wing disc cells for Dpp morphogen transference (*Roy et al., 2014*). ASP cells receive Dpp by emitting long cytonemes that contact with the target cell of the wing disc by their tips. This process was visualized by the same membrane-bound GRASP. Although Dpp uses a cytoneme-mediated contact-dependent mechanism for signalling at distance, the reception takes place where Dpp-receiving cytoneme tip contacts with the Dpp source cell body, in contrast with the cytoneme-cytoneme contacts here described for Hh. Nevertheless, the 'synaptic contacts' described for Dpp reception in ASP might be conceptually comparable with the 'kissing rings' distributed along the length of overlapping Hh-sending and Hh-receiving cytonemes.

We have previously demonstrated that Hh moves along cytonemes in multivesicular bodies

**Video 7.** Filopodia and *ptc* mutant region manual tracking using MTrackJ. (A-C) Abdominal histoblasts of a pupae with a *y w hs-Flipase122; FRT42D ptc^16 / FRT42D ubiRFPnls; hh.Gal4 / UAS.GMA-GFP* genotype. GMA-GFP expressed during all development and two days *ptc^16* clones (induced by heat-shock of 1 hr at 37°C). (A) Z-projection of GMA-GFP using the inverted grey-scale lookup table of ImageJ. (B) Z-projection of GMA-GFP and the tracking of filopodia using the MTrackJ plugin of Fiji, where the fist number refers to the number of filopodium tracked and the second name refers to the base (1) or the tip (2). Each filopodium has a colour and the coloured lines are the base and tip trajectories. Notice that also two dying larval epithelial cells are tracked (orange tracks #165.1 and #165.2). (C) Nuclear RFP (magenta) wild-type nuclei and *FRT42D ptc^16* mutant clone (absence of magenta) tracked using the first and last region where there is no RFP-positive nuclei just anterior to the A/P boundary (blue tracks #164.1 and #164.2). Histoblasts move down towards the dorsal midline, and this is not shown because of the high magnification. Anterior is on the left. Pupa is around 30 hr APF (after puparium formation). Movie of 30 min imaging with a time interval between frames of 30 s. Scale bars represent 10 µm.

(MVB) to be secreted as extracellular vesicles known as exosomes (*Gradilla et al., 2014*), which raises intriguing questions about the mechanism of interaction between exosomal Hh and its receptor Ptc in the A cytonemes. Interestingly, Ptc and Hh can be detected co-localizing with cytoneme contact sites, suggesting that they could represent specific sites for Hh reception and internalization. These data strengthen our proposal of cytoneme-mediated Hh morphogen distribution and suggest that Hh containing exosomes could be received at intimate contacts between P and A cytonemes.

Finally, we have shown that cytonemes are able to cross $ptc^{-/-}$ and $smo^{-/-}$ clones, enabling the Hh transport from P to A cells for Hh signalling reestablishment in wild type cells located anterior to the clones. In agreement, it is known that the Hh signal can cross small mutant clones of $ptc$ and $smo$ to reach the adjacent wild type territory. Similar behaviour, has been described for $ptc^{-/-}$ or $smo^{-/-}$ clones induced in the A abdominal histoblast nests and attributed to different cell polarity and cell affinity (*Struhl et al., 1997*). Therefore, Hh transport and reception by cytoneme-mediated cell-cell contacts can explain the Hh signalling in wild type cells located anterior to $ptc^{-/-}$ and $smo^{-/-}$ clones.

In summary, the data strengthen our proposal of cytoneme-mediated Hh morphogen distribution and suggest that Hh containing exosomes could be received at intimate contacts between P and A cytonemes. The described intimate interaction between these two sets of cytonemes at membrane contact sites, where ligand and receptor complex colocalize, may facilitate morphogen reception.

# Materials and methods

## Fly mutants

A description of mutations, insertions and transgenes is available at Fly Base (http://flybase.org). The following mutants and transgenic strains were used: *tub.Gal80ts, hs-Flp122* (Bloomington Drosophila Stock Centre (BDSC), Indiana, USA; http://flystocks.bio.indiana.edu), $hh^{ts2}$ (*Ma et al., 1993*), $shi^{ts1}$ (*Grigliatti et al., 1973*), $dally^{32}$ (*Franch-Marro et al., 2005*), $dlp^{20}$ (*Franch-Marro et al., 2005*), $smo^3$ (*Nüsslein-Volhard and Wieschaus, 1980*) and $ptc^{16}$ (*Nakano et al., 1989*).

## Overexpression experiments

The following *Gal4* and *LexA* drivers were used for ectopic expression experiments using the *Gal4/UAS* (*Brand and Perrimon, 1993*) and *LexA/LexAop* (*Yagi et al., 2010*) systems: *hh.Gal4* (*Tanimoto et al., 2000*), *ptc.Gal4* (*Hinz et al., 1994*), *dpp.LHG* (*Yagi et al., 2010*) (referred as *dpp.LexA*), *hh.LexA* (generated at CBMSO department of development and differentiation), *en.Gal4* (expressed exclusively in the P compartment was a gift from Christian Dahman).

The *pUAS*-transgene and *LexAop*-transgene strains were: *UAS.hh-GFP* (*Torroja et al., 2004*), *UAS.dlp-GFP* (*Han et al., 2004b*), *UAS.dally-GFP* (*Eugster et al., 2007*), *UAS.shi^{K44A}* (*Moline et al., 1999*), *UAS.ihog-YFP* and *UAS.ihog-RFP* (*Callejo et al., 2011*), *UAS.GMA-GFP* (*Bloor and Kiehart, 2001*), *UAS.CD4-tdTomato* (*Han et al., 2011*), *UAS.CD4-GFP^{1-10}* and *LexAop-CD4^{11}* (a gift from Kristin Scott), *UAS.hh-CD2* (*Strigini and Cohen, 1997*), *UAS.lifeactin-GFP* (BDSC 35544), *UAS.scar-RNAi* (BDSC 36121), *UAS.dia-RNAi* (BDSC 33424), *UAS.Cpa-RNAi* (Vienna Drosophila RNAi Center (VDRC), Vienna, Austria, v16731), *UAS.cher-RNAi* (BDSC 35755), *UAS.ptc-RNAi* (BDSC 28795), *UAS.smo-RNAi* (BDSC 27037), *UAS.ihog-RNAi* (VDRC 102602), *UAS.boi-RNAi* (VDRC 108265), *UAS.dally-RNAi* (VDRC 14136), *UAS.dlp-RNAi* (VDRC 106578).

The *LexAop.ihog-RFP* construct used *ihog-RFP* from the *pTWR* vector (*Callejo et al., 2011*), which was introduced into the *pLOTattB* plasmid (*Lai and Lee, 2006*) carrying the *lexA* operator (*LexAop*). Transgenic strains were recovered using standard protocols.

Transient expression of transgenic constructs used the *tub-Gal80ts; Gal4* and *LexA* systems with fly crosses maintained at 18°C, with inactivation of the *Gal80ts* repressor for 16–40 hr at restrictive temperature (29°C) before dissection. The transgene actin<CD2<Gal4 (*Pignoni and Zipursky, 1997*) was used to generate random ectopic clones of the *UAS* lines. Larvae of the corresponding genotypes were incubated at 37°C for 10 min to induce *hs-Flp*-mediated recombinant clones.

## Experimental genotypes and clonal analysis

Mutant clones were generated by *hs-Flp*-mediated mitotic recombination. Larvae were incubated at 37°C for 45 min at 48–72 hr after egg laying (AEL). The genotypes were:

*Figure 5A*: *y w hs-Flp122; FRT42D ptc[16] / FRT42D arm.lacZ.*

*Figure 5B*: *y w hs-Flp122; FRT42D ptc[16] / FRT42D arm.lacZ; UAS.ihog-RFP/ tub.Gal80[ts], hh.Gal4.*

*Figure 5C*: *y w hs-Flp122; FRT42D ptc[16] / FRT42D ubi.RFPnls; UAS.GMA-GFP / hh.Gal4.*

*Figure 5D,E*: *y w hs-Flp122; FRT42D ptc[16], UAS.CD4-GFP[1-10], LexAop.CD4-GFP[11]/ arm.lacZ FRT42D; dpp.LexA, tub.Gal80[ts] / LexAop.ihog-RFP, hh.Gal4.*

*Figure 6A*: *y w hs-Flp122; smo[3] FRT40A / arm.lacZ FRT40A.*

*Figure 6B*: *y w hs-Flp122; smo[3] FRT40A, ptc.Gal4 / arm.lacZ FRT40A; UAS.ihog-RFP / tub. Gal80[ts].*

*Figure 6C*: *y w hs-Flp122; smo[3] FRT40A / arm.lacZ FRT40A; UAS.ihog-RFP / tub.Gal80[ts], hh.Gal4.*

*Figure 6D*: *y w hs-Flp122; smo[3] FRT40A / arm.lacZ FRT40A, UAS.CD4-GFP[1-10], LexAop.CD4-GFP[11]; dpp.LexA, tub.Gal80[ts] / hh.Gal4.*

*Figure 7D*: *y w hs-Flp122; ptc.Gal4, tub.Gal80[ts] / UAS.ihogRFP; dally[32] dlp[20] FRT2A / ubi GFP FRT2A.*

*Video 6*: *y w hs-Flp122; FRT42D ptc[16] / FRT42D ubi.RFPnls; UAS.GMA-GFP / hh.Gal4.*

## Immunostaining of imaginal discs

Immunostaining was performed according to standard protocols (*Capdevila and Guerrero, 1994*). Imaginal discs from third instar larvae were fixed in 4% paraformaldehyde (PF) in PBS for 20 min at room temperature (RT) and permeabilized in PBS 0,1% Triton (PBT) before incubating with PBT 1% BSA for blocking (1 hr at RT) and primary antibody incubations (overnight at 4°C). Incubation with fluorescent secondary antibodies (1/400 ThermoFischer) was performed for 1 hr at RT and then washing and mounting in mounting media (Vectashield). Primary antibodies were used at the following dilutions: rabbit polyclonal anti-Hh (α-Hh) (*Bilioni et al., 2013*), 1:500; mouse monoclonal α-Ptc (*Capdevila and Guerrero, 1994*), 1:150; rat monoclonal α-Ci (a gift from B. Holmgren), 1:20, mouse monoclonal α-Dlp (*Lum et al., 2003*), 1:30; mouse monoclonal α-Smo (Hybridome bank), 1:30; rabbit polyclonal α-βGal (from Jackson laboratories), 1/1000.

The protocol for the extracellular labelling using α-Hh and α-Ptc antibodies is described in (*Torroja et al., 2004*). Imaginal discs from third instar larvae were dissected on ice, transferred immediately to ice-cold M3 medium containing α-Hh (1:30 dilution) and α-Ptc (1:10 dilution) antibodies and incubated at 4°C for 1 hr. The incubation with the primary antibody under these 'in vivo' conditions, without detergents prior to fixation, prevented antibody penetration of cells. Imaginal discs were then washed in ice-cold PBS, fixed in PBS 4% PF at 4°C, washed in PBT and incubated with secondary fluorescent antibody as above.

## Microscopy and image processing of imaginal discs

Laser scanning confocal microscope (LSM710 Zeiss) was used for confocal fluorescence imaging of imaginal discs. ImageJ software (National Institutes of Health) was used for image processing and for image analysis.

## Quantification of cytonemes and gradient extension in wing discs

Gradient lengths were determined as described in *Bischoff et al. (2013)*. To quantify the maximum extent of cytonemes, we measured the ten longest protrusions in the wing pouch, from the A/P border to the tip, using the line tool in ImageJ. Then, the average length of the ten longest cytonemes was used for the quantitative analysis.

## In vivo imaging of pupal abdominal histoblasts

Imaging of pupal abdominal histoblasts was done using a chamber as described in *Seijo-Barandiarán et al. (2015)*. Hh signalling filopodia from histoblasts of dorsal abdominal segment A2 were filmed using 40x magnification taking Z-stacks of around 30 μm of thickness with a step size of 1 μm every 2 min (*Video 1* and *Video 2*) or 30 s (*Video 6* and *Video 7*) using a LSM710 confocal microscope. All movies were analysed with Fiji and displayed at a rate of 7 frames per second. All imaged pupae developed into pharate adults and hatched normally.

## Manual tracking of signalling filopodia

Signalling filopodia were tracked using MTrackJ plugin of ImageJ (https://imagescience.org/meijering/software/mtrackj/). We took base (track#1) and tip (track#2) point (x, y) coordinates in each frame for each signalling filopodium (cluster) and coloured them by cluster, showing filopodia in different colours. Signalling filopodia were analysed in a 30 min time-window in all movies. We discarded filopodia not clearly visible within the 30 min movies due to tissue movement or surrounding dying larval epithelial cells (as shown in *Video 6*). The tracking was done in a region of 49 µm x 76 µm for experiments expressing GMA in Hh-producing (N = 4 pupae) or Hh-receiving (N = 4 pupae) cells, and for experiments co-expressing GMA and Ihog in Hh-producing (N = 4 pupae) or Hh-receiving (N = 4 pupae) cells. Tracking was restricted to a region of 79 µm x 116 µm for the $ptc^{-/-}$ mutant clone experiment (N = 1 pupa), where filopodia from the $ptc^{-/-}$ mutant territory and the wild-type territory was defined by the area references shown in *Video 7*, and used for statistical analysis.

## Data analysis of signalling filopodia dynamics

Using the (x, y) coordinates of base and tip points of each filopodium given by the MTrackJ we calculated the elongation velocity, retraction velocity and lifetime as we describe below in 'Filopodia dynamics models description'. We also calculated filopodium extent through this formula:

$$\text{Filopodium extent (E): } E = \sqrt{\left(x_{tip} - x_{base}\right)^2 + \left(y_{tip} - y_{base}\right)^2}$$

## Description of filopodia dynamic models

We have distinguished two populations of filopodial dynamic behaviours (*Figure 1—figure supplement 1*):

### Two phases triangle dynamics model

Filopodia dynamics can be described with two phases: elongation and retraction.

In this model, the base of the triangle is the filopodium lifetime; the elongation velocity (Ve) and the retraction velocity (Vr) correspond to the slopes of the triangle, that have been calculated from the experimental data connecting the filopodium origin and end with the maximum extent, respectively. The filopodium origin (and end) has been taken in the previous (next) frame of the recorded data.

### Three phases trapezoid dynamics model

Filopodia dynamics can be described with three phases: elongation, stationary and retraction.

In this model, the base is the filopodium lifetime, the slopes of the left and right sides of the trapezoid are the elongation velocity (Ve) and retraction velocity (Vr) respectively, which have been calculated respect to the stationary phase as follows:

The stationary phase is characterized by two main parameters: the average extent (Es) and average velocity (Vs), where Es is the filopodium extent at the half lifetime, and Vs is the average velocity of the velocities in the stationary phase.

The left trapezoid side is drawn by connecting the filopodium origin with the intersection point with Vs, which corresponds to the first experimental point equal or greater than Es. Equally, the right trapezoid side is drawn by connecting the filopodium end with the intersection point with Vs, which correspond to the last experimental point equal or greater Es.

## Statistical analysis of signalling filopodia dynamics

We used R (http://www.R-project.org) to perform the Shapiro-Wilk normality test and pooled data of the studied variables was compared between different genotypes using the Mann-Whitney-Wilcoxon test of homogeneity of variances.

For statistical analysis of Emax comparing Hh producing and receiving filopodia dynamics we used pooled data comparing 104 filopodia of 4 *ptc.Gal4>UAS.GMA-GFP* pupae, 100 filopodia of 4 *hh.Gal4>UAS.GMA-GFP* pupae, 47 filopodia of 4 *ptc.Gal4>UAS.GMA-GFP, UAS.ihog-RFP* pupae and 75 filopodia of 4 *hh.Gal4>UAS.GMA-GFP, UAS.ihog-RFP* pupae. Statistical analysis of 72, 79, 28 and 30 filopodia for Ve and 94, 82, 36 and 31 for Vr was done from *ptc.Gal4>UAS.GMA-GFP; hh. Gal4>UAS.GMA-GFP; ptc.Gal4>UAS.GMA-GFP, UAS.ihog-RFP;* and *hh.Gal4>UAS.GMA-GFP, UAS.*

*ihog-RFP* pupae, respectively; because not all filopodia had an elongation or retraction phase during the recording period.

For statistical analysis of Emax of Hh producing filopodia dynamics either crossing $ptc^{-/-}$ mutant or wild-type territories we compared 23 filopodia crossing $ptc^{-/-}$ mutant clone territory with 29 filopodia crossing wild-type territory of 1 pupa. For statistical analysis of Ve and Vr we used that data of 17 filopodia crossing wild-type territory and 14 filopodia crossing $ptc^{-/-}$ mutant territory in an elongation and a retraction phase.

## Acknowledgements

We are grateful to Mar Casado in our department for generating the LexA Hh insertion, to Eva Caminero for injecting the embryos to produce the transgenic fly lines and to the confocal microscopy facilities of the Centro de Biología Molecular 'Severo Ochoa' for skillful technical assistance. We are also grateful to the members of IG lab for discussions during the development of this work, to Marcus Bischoff for help with live imaging and to David Gubb, Pedro Ripoll, Ana-Citlali Gradilla and Adrián Aguirre-Tamaral for comments on the manuscript. We thank to Kristin Scott for the fly lines to do the GRASP experiments, to Xinhua Lin for the *dlp-GFP* and *dally-GFP* fly strains, to Christian Dahman for the *en.Gal4* expressed exclusively in the P compartment, and the BDSC and VDRC for stocks. Work was supported by grants BFU2014-59438-P from the Spanish Ministry of Economy and Competitiveness (MINECO), and by a personal and by institutional grants from the *Fundación Areces*. ISB were supported by FPI fellowship from the Spanish Ministry of Economy and Competitiveness (MINECO). L. González-Méndez was supported by a contract of BFU2014-59438-P grant from the Spanish MINECO).

## Additional information

### Funding

| Funder | Grant reference number | Author |
| --- | --- | --- |
| Ministerio de Economía y Competitividad | BFU2014-59438-P and BFU2015-72831-EXP | Isabel Guerrero |
| Fundación Ramón Areces | FRA14 | Isabel Guerrero |
| Ministerio de Economía y Competitividad | Graduate Student Fellowship (BFU2011-25987) | Isabel Guerrero |

The funders had no role in study design, data collection and interpretation, or the decision to submit the work for publication.

### Author contributions

LG-M, Conceptualization, Resources, Data curation, Formal analysis, Supervision, Validation, Investigation, Visualization, Methodology, Writing—original draft, Writing—review and editing; IS-B, Data curation, Validation, Investigation, Visualization, Methodology, Writing—review and editing; IG, Conceptualization, Resources, Data curation, Supervision, Funding acquisition, Validation, Investigation, Visualization, Writing—original draft, Project administration, Writing—review and editing

### Author ORCIDs

Isabel Guerrero, http://orcid.org/0000-0001-6761-1218

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
