## [Decision Letter]

Thank you for submitting your article "Cytoneme-mediated cell-cell contacts for Hedgehog reception" for consideration by *eLife*. Your article has been reviewed by three peer reviewers, and the evaluation has been overseen by a Reviewing Editor and Diethard Tautz as the Senior Editor. The following individuals involved in review of your submission have agreed to reveal their identity: Steffen Schlopp (Reviewer #2).

The reviewers have discussed the reviews with one another and the Reviewing Editor has drafted this decision to help you prepare a revised submission.

Summary:

This interesting work extends the study of filopodia/cytoneme-mediated signaling in a powerful invertebrate system. González-Méndez et al., take advantage of high-resolution imaging and genetics to elucidate the function of cytonemes in *Drosophila*. Here intercellular Hh trafficking is investigated with a particular focus on the receiving cells. The authors demonstrate that the receiving cells project signalling filopodia decorated with iHog, Ptc and Smo towards the Hh expressing compartment. Using GFP reconstitution experiments, they demonstrate that sending and receiving cytonemes come in close contact. Hh is delivered via cytonemes from the signalling cells to cytonemes of the receiving cell to induce signalling. Furthermore, the authors focus on gradient formation and show that, as previously reported for sending cells, cytonemes in receiving cells also play an important part in gradient formation.

Essential revisions:

1) Both reviewer #2 and #3 were not entirely convinced about the effects of the receiving cell cytonemes on Hh gradient formation and signaling output, and reviewer #1 also agreed that more data could be delivered to address this issue. However the reviewers didn't all concur that looking at Wg or Dpp signaling would be helpful. Considering this, we request that you provide additional data supporting your contention that receiving cell cytonemes affect gradient formation and Hh signaling output.

2) Most of data showing cytonemes were generated under conditions of Ihog overexpression. If such extensions indeed play important roles in transporting Hh, we would expect to see higher levels of Hh protein when Ihog is overexpressed in A or P, or both compartments. Such data is important to logically correlate cytoneme extensions with their roles in Hh transport and dispersion. Can you please provide data addressing this issue?

3) As a corollary to point 2 above, reviewers #1 and #2 would like to see additional data supporting a few key results in the absence of Ihog overexpression.

[Editors' note: further revisions were requested prior to acceptance, as described below.]

Thank you for resubmitting your work entitled "Cytoneme-mediated cell-cell contacts for Hedgehog reception" for further consideration at *eLife*. Your revised article has been favorably evaluated by Diethard Tautz (Senior editor), a Reviewing editor and two reviewers.

Both reviewers are quite enthusiastic about the manuscript, but one reviewer raised some questions that should ideally be addressed by further revisions. Please see the comments below:

1) The authors show that Ihog overexpression increases the lifetime of cytonemes and in previous work showed that the spread of Hh and Hh signaling correlate with cytoneme presence and length. Don't we expect that ptc.Gal4>ihog might increase Hh spread, Hh signaling and ptc expression? I may have missed it, but did the authors look at Ptc expression +/- ptc.Gal4>ihog?

2) Although it is highly unlikely that Ihog overexpression creates a new mechanism of signaling, the Figure 1 panels show Hh accumulation at both apical and basal surfaces of the P compartment cells. Is the accumulation at the basal surface a consequence of Ihog overexpression? If it is, it might relevant to the interpretations of the findings reported in the paper and to the relative importance of sending and receiving cytonemes. It would be helpful to show Hh in a disc that does not overexpress Ihog.

3) Figure 1' shows that the red and green fluorescence do not precisely coincide and the authors conclude that the red fluorescent Ihog and green fluorescent Hh are in distinct spaces and separate structures. It is my understanding that the resolution limit for confocal z sections is approximately 0.4–0.5microns, yet the diameter of a cytoneme is around 200nm and a synaptic gap is around 20–50nm. The authors do not indicate what objective was used or whether the images were processed with a deconvolution routine, so it is not obvious what the color separation in the confocal images might mean. The offset of red and green channels may be a technical artifact, namely chromatic aberration. Is there an independent way to define the spatial resolution in z with their microscope? It seems that the authors propose a model with three layers arranged in apical to basal order – the plasma membrane, receiving cytonemes, and sending cytonemes. If the plasma membrane of the P compartment cells is marked, does the same confocal imaging resolve Ihog-RFP basal to the cell membrane? Is there a way to show all three layers? In the proposed model, what are the minimum dimensions of the structures and what must their spatial separation be to be consistent with the images?

4) Subsection “Cytonemes from producing and receiving cells interact for Hh reception” "…these results clearly indicate that Hh reception is mediated by contacts between Hh-sending and Hh-receiving cytonemes." I would urge the authors to temper this statement to acknowledge that it is based on observations with Ihog over-expression and that Ihog overexpression may change the relative probabilities of the alternative modes of transfer by sending and receiving cytonemes.

5) Subsection “Descrete sites for cytoneme-mediated Hh reception” It would be appropriate to cite previous uses of GRASP to image long distant contacts and to contrast the results obtained here with previously published results.

6) Subsection” Cytonemes bridge *ptc-/-* and *smo-/-* mutant clones to re-establish Hh reception in adjacent territories” "Therefore, the presence of the network of cytonemes is independent of Hh signaling." The authors might consider being more cautious and to qualify this statement, as their conclusion is based on the effects they observed under conditions of partial Hh inactivation and Ihog over-expression."

We encourage you to address these points in a revision. It should be satisfactory to do this without adding new data. Point #4 is especially important to address, since if the color offset seen in Figure 1', and H" is actually a technical imaging artifact, then it is not biologically relevant and the interpretation of this figure must be changed. In your response to the reviews, and the revision, please note the resolution of your imaging system in Z and XY dimensions, please add scale bars where they are missing (Figure 1', H"), and please provide some evidence that chromatic aberrations have not confounded the interpretation of the results. This seems most likely to be an issue in the Z-axis imaging (Figure 1, Figure 3).

---

## [Author Response]

*Essential revisions:*

*1) Both reviewer #2 and #3 were not entirely convinced about the effects of the receiving cell cytonemes on Hh gradient formation and signaling output, and reviewer #1 also agreed that more data could be delivered to address this issue. However the reviewers didn't all concur that looking at Wg or Dpp signaling would be helpful. Considering this, we request that you provide additional data supporting your contention that receiving cell cytonemes affect gradient formation and Hh signaling output.*

We provide additional data to support the role of receiving cell cytonemes on Hh gradient formation and signaling output. We have included (in the new Figure 1) the expressions of the RNAis of SCAR and Dia in the receiving cells. We have observed that the knocking down of these proteins affects the cytoneme extension and leads to shorter extension of the *ptc* and *ci* gene expression domains.

*2) Most of data showing cytonemes were generated under conditions of Ihog overexpression. If such extensions indeed play important roles in transporting Hh, we would expect to see higher levels of Hh protein when Ihog is overexpressed in A or P, or both compartments. Such data is important to logically correlate cytoneme extensions with their roles in Hh transport and dispersion. Can you please provide data addressing this issue?*

We have previously demonstrated that Ihog overexpression in the P compartment cells increases Hh retention (Bilioni et al., 2013). This Hh retention in the P compartment cells decreases the extension of the Hh gradient (Bilioni et al., 2013). Here we show that Ihog overexpression in the A compartment cells also increases Hh sequestration (Figure 1’, G”) and changes the shape of the Hh gradient (causing an extended gradient, data not shown).

In this manuscript we have not shown the Hh gradient in conditions of Ihog overexpression in the A compartment. Data showing how the manipulation of actin dynamics affects Hh gradient was done without the overexpression of Ihog. The co-expression of Ihog was performed only to better visualize the effect on cytonemes of the expression of RNAis that knock down actin dynamics.

*3) As a corollary to point 2 above, reviewers 1 and 2 would like to see additional data supporting a few key results in the absence of Ihog overexpression.*

Although the best cytoneme marker is Ihog because of its effect on the stabilization of cytonemes, in the revised version we have included data using the expression of the actin-binding domain of moesin fused to GFP (GMA-GFP). Importantly, we demonstrate that the maximum extension of filopodia is similar with or without overexpression of Ihog (Figure 1, Figure 1—figure supplement 2). This overexpression, however, changes cytoneme dynamics as we previously showed in Bischoff et al., 2013. Moreover, the membrane-bound GRASP fluorescence (CD4-GFP) also shows the contacts along the membranes of interacting A and P cytonemes. And these experiments were done without Ihog overexpression (Figure 4, Video 5).

[Editors' note: further revisions were requested prior to acceptance, as described below.]

*Both reviewers are quite enthusiastic about the manuscript, but one reviewer raised some questions that should ideally be addressed by further revisions. Please see the comments below:*

*1) The authors show that Ihog overexpression increases the lifetime of cytonemes and in previous work showed that the spread of Hh and Hh signaling correlate with cytoneme presence and length. Don't we expect that ptc.Gal4>ihog might increase Hh spread, Hh signaling and ptc expression? I may have missed it, but did the authors look at Ptc expression +/- ptc.Gal4>ihog?*

Indeed ptc.Gal4>ihog increases Hh gradient extension, as shown by the expression of Ptc and Ci in the wing imaginal disc. To show these data we have included a new supplementary figure (Figure 1—figure supplement 3).

*2) Although it is highly unlikely that Ihog overexpression creates a new mechanism of signaling, the Figure 1 panels show Hh accumulation at both apical and basal surfaces of the P compartment cells. Is the accumulation at the basal surface a consequence of Ihog overexpression? If it is, it might relevant to the interpretations of the findings reported in the paper and to the relative importance of sending and receiving cytonemes. It would be helpful to show Hh in a disc that does not overexpress Ihog.*

Certainly, Ihog overexpression recruits Hh to cytonemes. However, there are other results described in the manuscript that support a cytoneme-mediated Hh reception in the A compartment cells that were obtained without Hh overexpression:

Figure 2 shows Hh accumulated in basal conduits resembling filopodia oriented to the A-P axis when Hh internalization is blocked (shi^DN^).

Figure 4 shows Hh accumulation in shi^DN^ cells that colocalizes at contact sites between A and P compartment cytonemes.

*3) Figure 1' shows that the red and green fluorescence do not precisely coincide and the authors conclude that the red fluorescent Ihog and green fluorescent Hh are in distinct spaces and separate structures. It is my understanding that the resolution limit for confocal z sections is approximately 0.4–0.5microns, yet the diameter of a cytoneme is around 200nm and a synaptic gap is around 20–50nm. The authors do not indicate what objective was used or whether the images were processed with a deconvolution routine, so it is not obvious what the color separation in the confocal images might mean. The offset of red and green channels may be a technical artifact, namely chromatic aberration. Is there an independent way to define the spatial resolution in z with their microscope? It seems that the authors propose a model with three layers arranged in apical to basal order – the plasma membrane, receiving cytonemes, and sending cytonemes. If the plasma membrane of the P compartment cells is marked, does the same confocal imaging resolve Ihog-RFP basal to the cell membrane? Is there a way to show all three layers? In the proposed model, what are the minimum dimensions of the structures and what must their spatial separation be to be consistent with the images?*

We do not fully understand the problem that the referee has with the images shown in Figure 1, since we do not describe signal colocalization in them. Due to the high resolution of the images, deconvolution was not needed for clearly distinguish the colour separation in confocal images, which means that these signals are in distinct slices of the Z-stacks. Regarding the chromatic aberrations, we did "dye-swap" and the result of the relative position of A and P compartment cytonemes was the same independently of colours of labelled cytonemes (see Figure 2’ and Figure 2’) (Figure 1’’). There is not an independent way to define the spatial resolution in z with our microscope. Figure 1 shows the plasma membrane of A compartment cells, and the A and P cytonemes. These cytonemes extend along the basal membranes, forming an intricate network. Due to the impossibility to specifically label cytonemes (we label the whole plasma membranes), we can only visualize individual cytonemes when they cross from one compartment to the other (not labelled). To determine the minimum dimensions of these GRASP labelled structures we need to do Electron Microscopy. The main message in this manuscript is to describe that interaction between P and A cytonemes mediates Hh reception in the A compartment cells. Additional work is needed to characterize the structures that mediate the Hh release and internalization.

Details of the images are the following:

Figure 1’

XY resolution = 14,45 pixel/mm

Z resolution = 16,38 pixel/mm

Number and dimension of slices = 9 slices of 0,253mm

Total Z-stack dimension = 2,28mm

Objective used: 100x

Figure 3

XY resolution = 9,11 pixel/mm

Z resolution = 9,09 pixel/mm

Number and dimension of slices = 8 slices of 0,303mm

Total Z-stack dimension = 2,42mm

Objective used: 100x

*4) Subsection “Cytonemes from producing and receiving cells interact for Hh reception” "these results clearly indicate that Hh reception is mediated by contacts between Hh-sending and Hh-receiving cytonemes." I would urge the authors to temper this statement to acknowledge that it is based on observations with Ihog over-expression and that Ihog overexpression may change the relative probabilities of the alternative modes of transfer by sending and receiving cytonemes.*

This statement is also based on observation in the absence of Ihog overexpression, as it is shown in Figure 1, Figure 2 and 3. Anyway, we have tempered our statement that Hh reception is mediated by contacts between Hh-sending and Hh-receiving cytonemes in the Discussion section of the revised manuscript.

*5) Subsection “Descrete sites for cytoneme-mediated Hh reception” It would be appropriate to cite previous uses of GRASP to image long distant contacts and to contrast the results obtained here with previously published results.*

As the reviewer requested, we have included the reference of a previous use of GRASP to imaging long distant contacts between non-neuronal cells (Roy et al., 2014).

*6) Subsection” Cytonemes bridge ptc-/- and smo-/- mutant clones to re-establish Hh reception in adjacent territories” "Therefore, the presence of the network of cytonemes is independent of Hh signaling." The authors might consider being more cautious and to qualify this statement, as their conclusion is based on the effects they observed under conditions of partial Hh inactivation and Ihog over-expression."*

The presence of cytonemes is confirmed in discs in which Hh function has been completely depleted (hh^ts^), as shown by the absence of Hh gradient included in Figure (Ci staining in Figure 5—figure supplement 2). Also, cytonemes are visualized with both Ihog over-expression (Figure 5—figure supplement 2), and the actin markers lifeactin-GFP (Figure 5—figure supplement 2) and GMA-GFP (Figure 5 and Video 6).

*We encourage you to address these points in a revision. It should be satisfactory to do this without adding new data. Point #4 is especially important to address, since if the color offset seen in Figure 1', and H" is actually a technical imaging artifact, then it is not biologically relevant and the interpretation of this figure must be changed. In your response to the reviews, and the revision, please note the resolution of your imaging system in Z and XY dimensions, please add scale bars where they are missing (Figure 1', H"), and please provide some evidence that chromatic aberrations have not confounded the interpretation of the results. This seems most likely to be an issue in the Z-axis imaging (Figure 1, Figure 3).*

We have addressed all the points suggested by the referee. Regarding point 4 we do not believe that data shown in Figure 1' and H" can be a technical imaging artefact. We asked our confocal microscopy facility specialists and they agree that these results cannot be technical imaging artefacts. The scale bars are now included in Figure 1', H" and 3B. The number and dimension of slices are included in Figure 1 and Figure 3 legends.

In the new version of the manuscript we have also included a new supplementary figure (Figure 1—figure supplement 3) to address point 1 of the reviewer. In addition, we have modified the text according to his/her suggestions.